# Controlled manipulation of oxygen vacancies using nanoscale flexoelectricity

Saikat Das[1,2], Bo Wang[3], Ye Cao[4,5], Myung Rae Cho [1,2], Yeong Jae Shin[1,2], Sang Mo Yang[4,6], Lingfei Wang[1,2], Minu Kim [1,2], Sergei V. Kalinin [4,5], Long-Qing Chen[3] & Tae Won Noh[1,2]

Oxygen vacancies, especially their distribution, are directly coupled to the electromagnetic properties of oxides and related emergent functionalities that have implications for device applications. Here using a homoepitaxial strontium titanate thin film, we demonstrate a controlled manipulation of the oxygen vacancy distribution using the mechanical force from a scanning probe microscope tip. By combining Kelvin probe force microscopy imaging and phase-field simulations, we show that oxygen vacancies can move under a stress-gradient-induced depolarisation field. When tailored, this nanoscale flexoelectric effect enables a controlled spatial modulation. In motion, the scanning probe tip thereby deterministically reconfigures the spatial distribution of vacancies. The ability to locally manipulate oxygen vacancies on-demand provides a tool for the exploration of mesoscale quantum phenomena and engineering multifunctional oxide devices.

---

[1] Center for Correlated Electron Systems, Institute for Basic Science (IBS), Seoul 08826, Republic of Korea. [2] Department of Physics and Astronomy, Seoul National University (SNU), Seoul 08826, Republic of Korea. [3] Department of Materials Science and Engineering, The Pennsylvania State University, Pennsylvania, PA 16802, USA. [4] Center for Nanophase Materials Sciences, Oak Ridge National Laboratory, Oak Ridge, TN 37831, USA. [5] Institute for Functional Imaging of Materials, Oak Ridge National Laboratory, Oak Ridge, TN 37831, USA. [6] Department of Physics, Sookmyung Women's University, Seoul 04310, Republic of Korea. Correspondence and requests for materials should be addressed to S.D. (email: saikat.das87@gmail.com) or to T.W.N. (email: twnoh@snu.ac.kr)

O xygen vacancies ($V_o^{..}$) are elemental point defects in oxides, and they generally function as mobile electron donors. At sufficiently high concentrations, they can disturb the ground state and promote emergent functional phenomena, such as superconductivity[1], ferromagnetism[2, 3], metal-to-insulator transition[4] and interface conductivity[5]. In addition, the distribution and dynamics of $V_o^{..}$ play a central role in many oxide-based energy and memory applications[6, 7]. Hence, the ability to manipulate $V_o^{..}$ provides an opportunity to study the evolution of emergent phenomena and control numerous functionalities, which are essential for developing next generation oxide devices[8, 9].

Traditionally, modification of the vacancy concentration has been carried out by annealing oxides at high temperature in reducing/oxidising atmosphere. Recent works have shown that at room temperature, the vacancy concentration can instead be locally changed by employing an electrical bias from a scanning probe microscope (SPM) tip[4, 10]. Such local manipulation enables reversible nanoscale control of metal-insulator transitions[4] and the modulation of interface conductivity[11]. However, in addition to the change in the vacancy concentration, the application of bias through an SPM tip is often accompanied by charge injection[12, 13] and the formation of protons/hydroxyls[14], which complicates the practical implementation of this approach.

Recently, it has been shown that the mechanical force from the SPM tip can also deplete $V_o^{..}$ from the contact region[15, 16]. However, because this mechanical depletion is less understood, the feasibility of using the force from an SPM tip as an active means of manipulating $V_o^{..}$ has not been explored. An explanation for the mechanical depletion of $V_o^{..}$ has been proposed through the so-called piezochemical coupling mechanism[15–17], which involves the converse Vegard effect and the flexoelectric effect. The converse Vegard effect accounts for the decrease in vacancy concentration due to force-induced lattice compression. Meanwhile, the flexoelectric effect considers the electromigration of positively charged $V_o^{..}$ under the stress-gradient-induced flexoelectric field. Notably, by definition, the flexoelectric effect refers to the generation of an electrical polarisation by stress-gradients[18, 19]. In this context, Yudin and Tagantsev suggested that the flexoelectric field is not a macroscopic electric field[19]; instead, it should be understood as a pseudo-internal field that polarises a medium in the presence of a stress-gradient[20–23]. Hence, it remains unclear how this flexoelectric field acts on $V_o^{..}$.

To be able to mechanically manipulate $V_o^{..}$, we must, therefore, thoroughly understand how they respond to the mechanical force from the SPM tip. Subsequently, we have to devise a strategy to employ this force to increase as well as decrease the vacancy concentration in a controlled manner.

Here, we demonstrate a controlled manipulation of $V_o^{..}$ using the mechanical force from an SPM tip. As a model system, we used a homoepitaxial thin film of $SrTiO_3$, which is an archetypal quantum paraelectric oxide with well-known flexoelectric coefficients[24, 25]. Using a Kelvin probe force microscopy (KPFM)-based imaging scheme, we show that besides pushing $V_o^{..}$ away in the vertical direction, the force promotes the lateral transport of vacancies during the movement of the tip. Using phase-field simulations, we argue that this mechanical redistribution of $V_o^{..}$ is driven by the depolarisation field associated with the stress-gradient-induced flexoelectric polarisation. The depolarisation field underneath the tip promotes the vertical migration of vacancies, whereas around the contact edge, it traps $V_o^{..}$ that can move with the tip. Furthermore, we demonstrate that altering the tip geometry can tailor this depolarisation field to preferentially allow the lateral transport of $V_o^{..}$. This approach enables a controlled spatial modulation of vacancies.

## Results

**Probing oxygen vacancies with Kelvin probe force microscopy.** We start by discussing the concept of probing distribution of $V_o^{..}$ with the KPFM technique (Fig. 1a–c). KPFM is a surface sensitive technique that measures the contact potential difference (CPD) between the tip and $SrTiO_3$ (STO) surface. The CPD is defined as the offset in the respective vacuum energy levels (in units of V)[26]. However, as schematically illustrated in Fig. 1b, this CPD would change if the STO surface contains $V_o^{..}$, which can be locally accumulated by an electrical poling. This change in the CPD can originate from the workfunction of STO[27], chemical dipoles (including $V_o^{..}$-electron pairs), and their orientation[28]. Moreover, the tip bias used during the KPFM measurement has been argued to influence the measured CPD[29]. The contribution of each factor cannot be individually separated, which inhibits a quantitative determination of the vacancy concentration with the KPFM technique. Nonetheless, as shown in Fig. 1c, the KPFM contrast across this $V_o^{..}$-rich region can be argued to scale proportionally with the vacancy concentration[27]. To establish a proof of concept, in the following we elaborate on the application of KPFM technique to study the diffusion of $V_o^{..}$ using a 14 and 120-unit cell (uc)-thick STO film (details of the film are provided in the Methods section and in Supplementary Fig. 1).

Figure 1d, e show KPFM images after poling the pristine surface of STO films with a tip bias of −5 V. The images of the 14-uc (120-uc)-thick STO film were taken 20 (16) minutes and 380 (360) minutes after poling. An image contrast is clearly visible across the poled region, which is appearing as a dark rectangular patch. This implies surface charging, either by injected charges, protons/hydroxyls, and/or $V_o^{..}$[30, 31]. The injected charges and protons/hydroxyls are expected to decay with a timescale, which is independent of the STO thickness[31]. In contrast, because of diffusion or surface reaction that enables the recombination of $V_o^{..}$ with oxygen from the ambient, the vacancy concentration is expected to decrease with a pronounced thickness-dependent timescale[32]. Figure 1d, e indeed show that the contrast across the poled region diminishes over time, and the decrease is largest in the 14-uc-thick STO film. Therefore, we argue that the surface charging is caused by $V_o^{..}$, which either undergo diffusion or recombine with oxygen from the ambient.

We can distinguish the dominating mechanism that causes the vacancy concentration to decay over time by analysing the time-dependency of the degree of equilibrium, $S(t)$ that can be calculated from the KPFM images (see Supplementary Note 1). The time-dependency of $S(t)$ describes how the surface equilibrates after the electrical poling. $S(t)$ will exhibit a semi-parabolic (linear) time-dependency if diffusion (surface reaction) is the dominating mechanism[27, 32]. Figure 1f plots $S(t)$ as a function of time. Evidently, the time-dependencies of $S(t)$ are semi-parabolic for both films–implying that the diffusion of $V_o^{..}$ causes the vacancy concentration to decrease over time.

Arguably, during the poling, the applied tip bias perturbs the equilibrium $V_o^{..}$-distribution of the entire film including the surface, which equilibrates through the diffusion of $V_o^{..}$. This diffusion occurs both along the out-of-plane and in-plane directions. However, due to the high surface sensitivity of the KPFM technique, the surface-bulk diffusion of $V_o^{..}$ predominantly affects the time evolution of the KPFM contrast, and thus the time-dependency of $S(t)$. Notably, during this surface-bulk diffusion, the repulsive vacancy–vacancy interaction inhibit $V_o^{..}$ to migrate independently along the out-of-plane direction[33]. Thus, the time-dependency of $S(t)$ effectively describes how the perturbed volume under the poled area (in Fig. 1d, e) equilibrates (see Supplementary Note 1 for a detailed discussion). Naturally, this volume would be smaller in the thinner STO film, and thus would equilibrate faster. This explains why the KPFM contrast

$(S(t))$ diminishes (grows) more rapidly in the 14-uc thick STO than in the 120-uc thick STO film (Fig. 1d–f).

Following the rationale above, we fit the time evolution of $S(t)$ with Fick's 2nd law of diffusion (solid lines in Fig. 1f). From this fitting, we obtained the diffusion coefficient $D = 9.4(3) \times 10^{-19}$ and $3.8(2) \times 10^{-18}$ cm$^2$ s$^{-1}$ for the 14-uc and 120-uc-thick STO film, respectively. These values are well in the range of the bulk value $D_{bulk} \approx 10^{-(17\pm3)}$ cm$^2$ s$^{-1}$ (300 K-extrapolated)[34], which validates the conceptual schematic depicted in Fig. 1c. Subsequently, we utilised this correlation between the vacancy concentration and KPFM signal to image the vacancy redistribution under an applied bias and force.

**Oxygen vacancy redistribution under applied bias and force.** In this section using the 120-uc-thick STO film, we compare the response of $V_o^{\cdot\cdot}$ to the electrical bias and force from the SPM tip. For this study, we employed a sharp tip with a radius of curvature = 25 nm. As depicted in the KPFM image (Fig. 2a), we scanned the left side of a formerly $V_o^{\cdot\cdot}$-enriched region with an increasingly positive tip bias (0–5 V) at a contact force = 0.6 μN. Meanwhile, the right side was scanned with an increasing contact force (0.6–8.5 μN) by holding the tip at the ground potential. In either case, the tip crossed the border between the formerly

$V_o^{\cdot\cdot}$-enriched and pristine regions. The central part of the KPFM image is uniformly dark, which indicates a uniform vacancy distribution. However, a change in the image contrast, which stems from the $V_o^{\cdot\cdot}$-redistribution is visible within both electrically and mechanically scanned areas. Notably, these scans together with acquiring the KPFM image took about 20 min. Because of their ultra-slow decay as evident from Fig. 1e, during this time $V_o^{\cdot\cdot}$ can be assumed to be kinetically frozen. This implies that the $V_o^{\cdot\cdot}$-redistribution is due to the applied bias and force.

To quantify this $V_o^{\cdot\cdot}$-redistribution, we constructed the vacancy concentration map from Fig. 2a by defining the normalised vacancy concentration, NVC $= \frac{(V_b - V)}{(V_b - V_{min})}$. Here, $V_b$ and $V_{min}$ represent the baseline and minimum KPFM signals, which were extracted from areas indicated by the black (top-right corner) and white (centre) squares, respectively. The reconstructed NVC map is shown in Fig. 2b. The positive (negative) NVC refers to a higher (lower) vacacny concentraion compared to the intrinsic surface concentration of $V_o^{\cdot\cdot}$ (NVC = 0, at the top-right corner).

Profiling the NVC map along lines E and M indicates that both the applied bias and force depleted the formerly $V_o^{\cdot\cdot}$-enriched region (Fig. 2c). We find that this depletion can be modelled with a phenomenological Boltzmann sigmoid function in the following

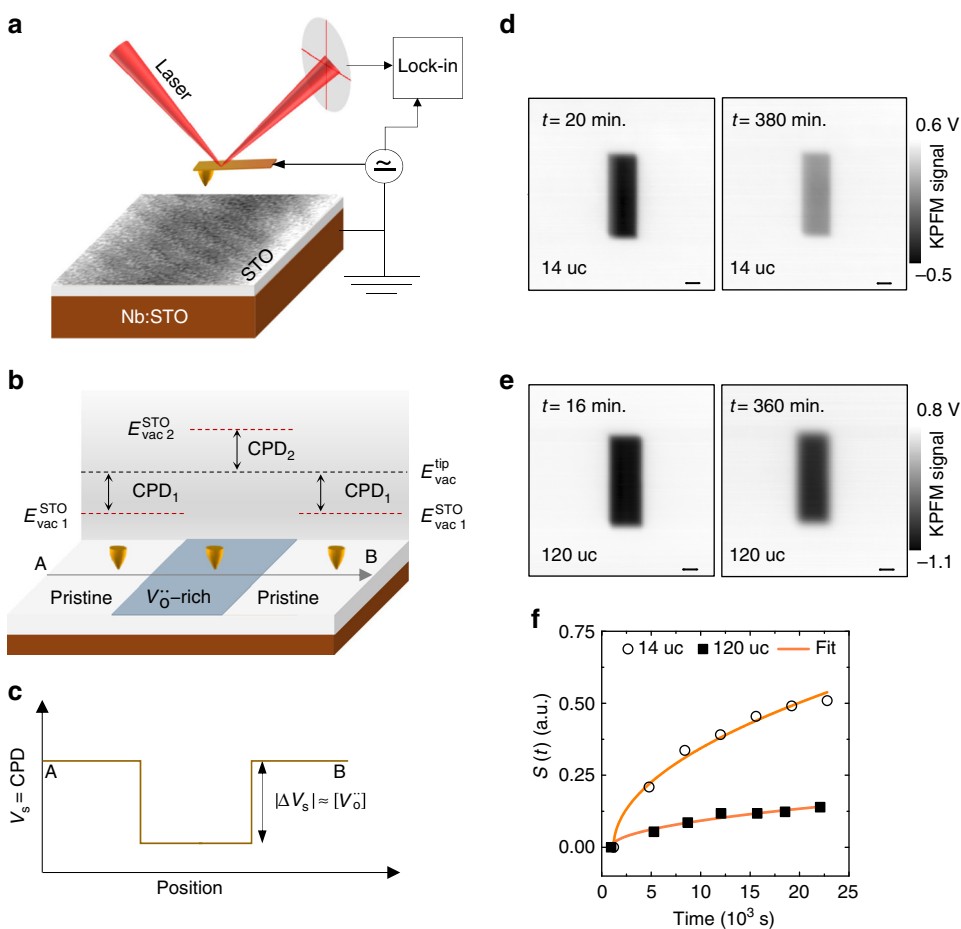

**Fig. 1** Studying diffusion characteristics of oxygen vacancies with KPFM. **a** Sketch of the sample geometry and KPFM measurement architecture. **b** Schematic illustration of the contact potential difference (CPD) contrast between the pristine (referred to as 1) and $V_o^{\cdot\cdot}$-rich (referred to as 2) regions. $E_{vac}$ denotes the vacuum energy level. **c** Illustrative KPFM signal from a line scan from position A to B across this $V_o^{\cdot\cdot}$-rich region. $\Delta V_s$ and $[V_o^{\cdot\cdot}]$ denote the net change in the measured KPFM signal and the concentration of $V_o^{\cdot\cdot}$ within the $V_o^{\cdot\cdot}$-rich region, respectively. **d**-**f** Characterisation of diffusion of $V_o^{\cdot\cdot}$ with the KPFM technique. KPFM images around a $V_o^{\cdot\cdot}$-enriched surface region of a 14-uc-thick **d** and 120-uc-thick **e** STO films. The time lag between poling the pristine surface and the time of acquiring an image is indicated on the *top-right corner* of KPFM image. The time evolution of the degree of equilibrium, $S(t)$ (*solid symbols*) and fit (*solid line*) according to Fick's 2nd law of diffusion **f**. The *scale bar* in **d**, **e** represents 1 μm

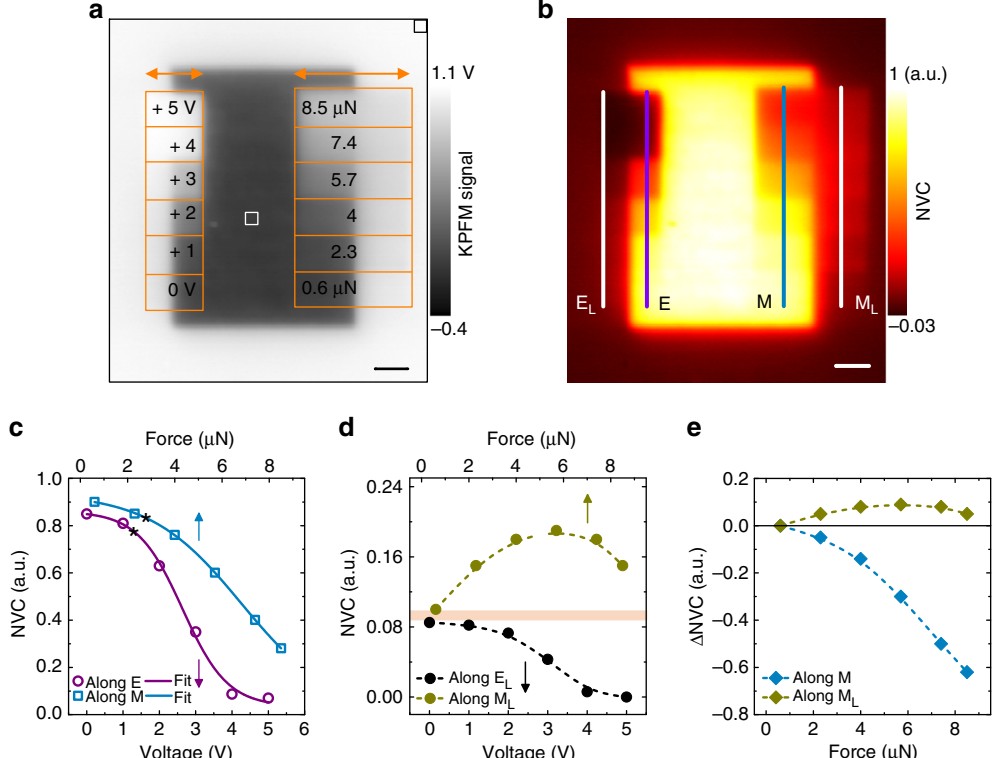

**Fig. 2** Characterisation of oxygen vacancy redistribution by applied bias and force. **a** The KPFM image after electrical and mechanical scans were performed across borders between the $V_o^{..}$-enriched and pristine regions. The schemes for electrical and mechanical scans are overlaid on the image. *Horizontal arrows* mark the corresponding fast scan direction. Before electrical and mechanical sans, the $V_o^{..}$-enrichment was performed by poling a $5 \times 7\ \mu m^2$ area of the pristine surface with a tip bias of −5 V. **b** The normalised vacancy concentration (NVC) map constructed from the KPFM image in **a**. **c** The NVC along lines E (*open circles*) and M (*open squares*) in **b** measured as a function of applied bias and contact force, respectively. The *solid lines* denote the Boltzmann sigmoid fit to the data, and Table 1 lists the corresponding best-fit parameters. The threshold voltage ($V_{th}$) and force ($F_{th}$) for the depletion of $V_o^{..}$ are marked with stars. **d** The NVC along lines $E_L$ (*black coloured circles*) and $M_L$ (*dark yellow coloured circles*) in **b** measured as a function of applied bias and contact force, respectively. The *dashed lines* are a guide for eyes. Lines $E_L$ and $M_L$ are placed 1 μm away from the borders between the $V_o^{..}$-enriched and pristine regions. The *thick horizontal line* in **d** marks the background along these lines. **e** Background-subtracted NVC (ΔNVC) along lines M (*turquoise coloured diamonds*) and $M_L$ (*dark yellow coloured diamonds*) in **b**. The *dashed lines* are a guide for eyes. NVC at 0.6 μN is used as the background. The data plotted in **c**, **d** are extracted using a 0.8 μm-wide averaging window. The *scale bar* in **a**, **b** represents 1 μm

**Table 1 Best-fit parameters from the Boltzmann sigmoid fit to the normalised vacancy concentration (NVC) data in Fig. 2c**

|  | $A_1$ | $A_2$ | $X_o$ | $M$ |
|---|---|---|---|---|
| Electrical | 0.86 (3) | 0.03 (4) | 2.6 (1) V | 0.6 (1) |
| Mechanical | 0.93 (1) | 0.004 (25) | 6.8 (1) μN | 1.9 (1) |

form,

$$f(x) = A_2 + \frac{A_1 - A_2}{\left[1 + e^{\frac{x - X_o}{M}}\right]} \quad (1)$$

where $A_1$, $A_2$, $X_o$, and $M$ refer to the initial value, final value, center, and decay rate, respectively. The *solid lines* in Fig. 2c are fits to this function. Table 1 lists the best-fit parameters. This functional analysis suggests that the applied positive bias and force have a same qualitative effect along the lines E and M, respectively. By comparing the $X_o$ ($= 0.5A_1$) values from the fits, we obtain a force-voltage equivalence factor of $0.4\ V\ \mu N^{-1}$. The Boltzmann sigmoid fits in Fig. 2c also exhibit an onset. Defining this onset as $0.9A_1$ (indicated by stars), on an ad hoc basis, yields a threshold voltage ($V_{th}$) and force ($F_{th}$) of 1.3 V and 2.7 μN or

equivalently 1.1 V, respectively. These threshold values are consistent with the activation barrier potential ($= 1$–$1.4$ V) for the electromigration of $V_o^{..}$ in STO[35, 36], which further validates the force-voltage equivalence relationship.

Surprisingly, electrical and mechanical scans yield the opposite image contrast across borders in the NVC map. As shown in Fig. 2d, along line $E_L$, NVC monotonically decreases with the increasing bias. However, along line $M_L$, NVC varies non-monotonously: first, it increases as the force increases and then gradually drops; the *horizontal line* in Fig. 2d marks the background (NVC = 0.1 at 0 V and 0.6 μN). We, therefore, conclude that the electrical scan depletes the pristine surface outside the left border, while the mechanical scan enriches the pristine surface outside the right border.

The force-induced vacancy-enrichment of the pristine region implies that some of the depleted $V_o^{..}$ moved laterally with the tip across the border during the mechanical scan. Additional experiments elaborate that the mechanical scan does not alter the background within the pristine region, and thus rule out the formation of $V_o^{..}$ and triboelectric charging of the STO surface during the mechanical scan (see Supplementary Note 2). To quantify the lateral motion of $V_o^{..}$, in Fig. 2e we, therefore, show the NVC with the background (NVC at 0.6 μN) subtracted (i.e., ΔNVC) along lines M and $M_L$. Because KPFM is a surface-sensitive technique, the remnant $V_o^{..}$ on the surface contribute to

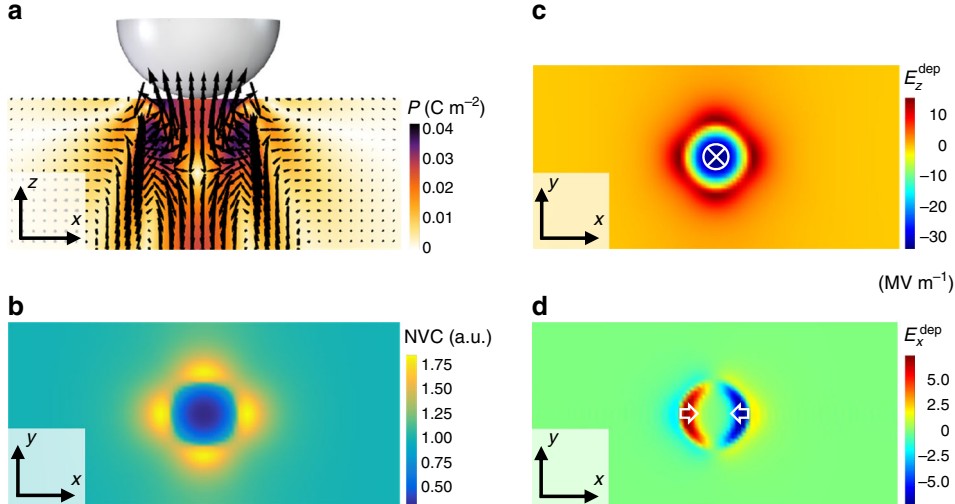

**Fig. 3** Phase-field simulations. **a** Simulated out-of-plane (z-x plane) vector map of flexoelectric polarisation induced by the scanning probe microscope (SPM) tip under a static contact force of 4 μN. *Arrows* denote the direction of the polarisation vectors, and their lengths correspond to the magnitude of the induced polarisation. **b** Simulated in-plane (x-y plane) normalised vacancy concentration (NVC) map around the tip-STO contact region, which shows a depletion and enrichment of $V_o^{\cdot\cdot}$ underneath the tip and around the contact edge, respectively. **c**, **d** The component-resolved in-plane distribution of depolarisation field around the tip-STO contact region. The z-component, $E_z^{dep}$ **c**, and the x-component, $E_x^{dep}$ **d**. The in-plane distribution of the y-component, $E_y^{dep}$, should be viewed as the same as the one in **d** but rotated by 90° in the x-y plane

the NVC map after $V_o^{\cdot\cdot}$-redistribution. Also, the possibility of recombination with oxygen from the ambient during the lateral motion of $V_o^{\cdot\cdot}$ can be ignored based on the following considerations. The influence of the surface reaction process, which could facilitate this recombination is negligible in our film. Furthermore, the use of a grounded tip during the mechanical scan rules out the bias-induced amplification of this surface reaction process[37]. Hence, for a particular force, the ratio between the net gain (along $M_L$) and the net drop in NVC (along M) represents the fraction of depleted $V_o^{\cdot\cdot}$ that moved laterally with the tip. By comparing the maximum net gain ( = 0.1), and corresponding drop ( = 0.3) at force = 5.7 μN, we estimated that only approximately 1/3 of the depleted $V_o^{\cdot\cdot}$ moved with the tip along the surface, while the rest migrate into the bulk.

Overall, our main observations are as follows. First, the contact force is bifunctional: it depletes the formerly $V_o^{\cdot\cdot}$-enriched region and simultaneously enriches the pristine surface. Second, the mechanical $V_o^{\cdot\cdot}$-redistribution involves a predominant surface-bulk migration and a relatively weaker lateral motion of $V_o^{\cdot\cdot}$ with the tip.

**Modelling the mechanical redistribution of oxygen vacancies.** To understand the mechanical redistribution of $V_o^{\cdot\cdot}$ we first considered two mechanisms–the converse Vegard effect and the flexoelectric effect[15, 16]. Recently, the magnitude of these two effects under an applied force from SPM tip has been compared in a PbTiO₃ (PTO) thin film[38]. This study suggests that the converse Vegard effect is much weaker than the flexoelectric effect. Notably, both the PTO and STO have comparable flexoelectric and Vegard coefficients[25, 38, 39], which determine the relative contributions of these two effects for a given force. Based on these considerations, we thus conclude that the flexoelectric effect predominantly causes the mechanical redistribution of $V_o^{\cdot\cdot}$, and the contribution from the converse Vegard effect is marginal.

For gaining a mechanistic understanding of this flexoelectric effect-driven $V_o^{\cdot\cdot}$-redistribution, we performed phase-field simulations; whereby we incorporated the flexoelectric effect[40] and coupled the time-dependent Ginzburg-Landau and the Nernst-

Planck equations[41, 42]. The simulation was performed assuming that the STO is paraelectric (see Supplementary Note 3). Unlike in the experiment, the SPM tip was assumed to be static, and following the Hertzian model the contact radius was calculated to be 8 nm for a contact force of 4 μN. Initially, $V_o^{\cdot\cdot}$ were assumed to be homogeneously distributed over the entire STO thickness, instead of being localised on the surface. Despite these oversimplified assumptions, our simulation still provides a qualitative insight into how $V_o^{\cdot\cdot}$ respond to mechanical stimuli. A detailed explanation of our model and a discussion on the redistribution of $V_o^{\cdot\cdot}$ under a positive tip bias are included in Supplementary Notes 4, 5.

The stress-gradient from the SPM tip locally polarises STO through the flexoelectric effect[24], as evident from the out-of-plane polarisation vector map in Fig. 3a. Since in the simulation the STO film is assumed to be in the paraelectric phase, the polarisation in Fig. 3a purely stems from the flexoelectric effect. This flexoelectric polarisation reaches a maximum ( ~ 0.04 C m⁻²) underneath the tip and spatially varies both in magnitude and direction. The resulting polarisation bound charge and the associated depolarisation field accordingly redistribute vacancies around the tip-STO contact region. This redistribution can be visualised from the simulated NVC map in Fig. 3b, which plots the in-plane distribution of $V_o^{\cdot\cdot}$ at the surface. Clearly, the vacancy concentration is decreased (enhanced) underneath the tip (around the contact edge).

An intuitive understanding of the simulated $V_o^{\cdot\cdot}$-redistribution can be gained from the component-resolved distribution of the depolarisation field. As shown in Fig. 3c, the z-component, $E_z^{dep}$, points downward (upward) below the tip (around the contact edge), whereas the x-component, $E_x^{dep}$, exhibits a parentheses-like structure: a node at the contact point and antinodes around the contact edge (Fig. 3d). Effectively, $E_x^{dep}$ points inwards, as indicated by the white arrows in Fig. 3d. The y-component (not shown), $E_y^{dep}$, forms an analogous structure to $E_x^{dep}$ but is rotated by 90° in the x-y plane. The decrease in the vacancy concentration underneath the tip can be attributed to the downward $E_z^{dep}$ component, which moves positively charged $V_o^{\cdot\cdot}$ from the surface into the bulk. In contrast, the combination of the upward $E_z^{dep}$ and inwardly directed $E_{x,y}^{dep}$ components favours the accumulation

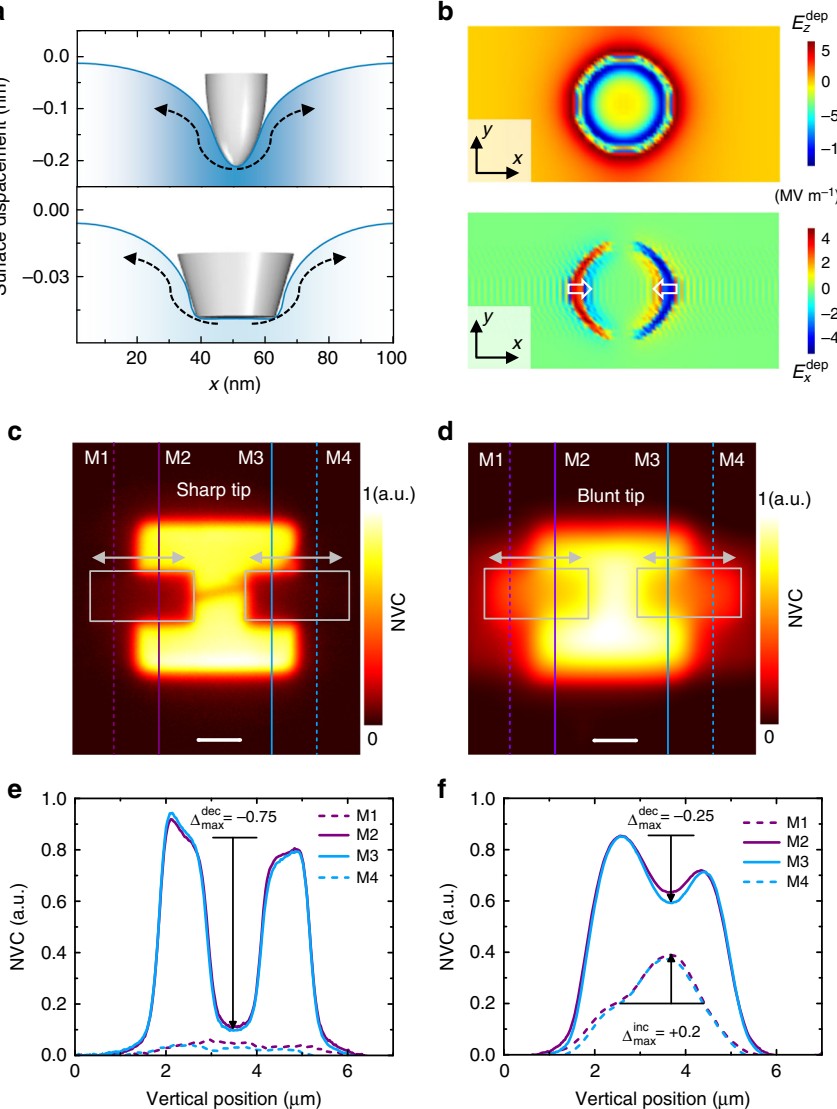

**Fig. 4** Controlled manipulation of oxygen vacancies. **a** Simulated surface deformation profiles under a spherical (*upper panel*) and flat-ended (*lower panel*) tip for a static contact force of 4 μN. **b** Simulated in-plane distribution of the *z*-component, $E_z^{dep}$ (*upper panel*) and *x*-component, $E_x^{dep}$ (*lower panel*) of the depolarisation field induced by the flat-ended tip. The ripples in $E_x^{dep}$ are numerical artefacts. **c, d** The normalised vacancy concentration (NVC) maps after mechanical scans were performed using a sharp **c** and blunt tip **d** with a contact force of 9.5 μN within the *grey coloured boxes*. *Horizontal arrows* mark the corresponding fast scan direction. Before mechanical sans, the $V_o^{..}$-enrichment were performed by poling the pristine surface with a tip bias of −5 V. **e, f** NVC profiles along lines M1, M2, M3 and M4 in **c**(**e**) and in **d**(**f**). M1 and M4 are placed 0.5 μm away from the borders between the $V_o^{..}$-enriched and pristine regions. The *vertical arrow* marks the maximum net increase ($\Delta_{max}^{inc}$) or decrease ($\Delta_{max}^{dec}$) in NVC. *Horizontal black lines* in **e**, **f** mark the background, which is used to estimate the net change in the NVC. The NVC profiles are averaged over a 0.5-μm-wide averaging window. Note that the boundaries between the $V_o^{..}$-enriched and pristine regions in **d** are more diffused compared to those in **c**. This is caused by the use of the blunt tip during the KPFM imaging. The *scale bar* in **c**, **d** represents 1 μm

of $V_o^{..}$ around the contact edge, yielding an increase in the vacancy concentration.

The simulation results qualitatively explain the characteristics of the mechanical $V_o^{..}$-redistribution in Fig. 2. During a mechanical scan, a spatially extended and strong downward $E_z^{dep}$ field acts over a larger fraction of $V_o^{..}$ underneath the tip, which results in a dominant surface-bulk migration. In contrast, a small fraction of $V_o^{..}$ becomes effectively trapped within a shallow annular region around the contact edge by the upward $E_z^{dep}$ and inward $E_{x,y}^{dep}$. These trapped vacancies can move laterally with the tip from the $V_o^{..}$-enriched region to the pristine region during the tip's lateral motion. Therefore, the contrasting roles of the depolarisation field underneath the tip and around the contact

edge corroborate both the dominant surface-bulk migration and the relatively weaker lateral motion of $V_o^{..}$ with the tip along the surface.

Notably, while scanning, the tip redistributes $V_o^{..}$ regardless whether it moves from left-right (trace) or right-left (retrace). Thus, the mechanical redistribution of $V_o^{..}$ should be understood as an average response of $V_o^{..}$ to the force applied during the trace and retrace. Moreover, the scanning velocity would influence the $V_o^{..}$-redistribution–longer the tip spends in contact with STO, the larger number of $V_o^{..}$ it would redistribute. To check whether these factors could contribute to the weaker lateral motion of $V_o^{..}$, we performed additional experiments, whereby we applied force only during the trace, and varied the scanning velocity

(see Supplementary Figs. 9–11). These experiments also yielded a weaker lateral motion but a stronger surface-bulk migration of $V_o^{..}$–highlighting the dominating influence of depolarisation field underneath the tip. In the following, we illustrate that the depolarisation field around the tip-STO contact junction can be tailored in favour of the lateral motion of $V_o^{..}$, which enables controllably manipulating the vacancy distribution.

**Controlled spatial modulation of oxygen vacancies**. The basic concept of tailoring the SPM tip-induced depolarisation field can be understood with the aid of Fig. 4a, which compares the simulated surface deformation profiles under a static load of 4 μN using two different tip geometries. The *upper panel* corresponds to the spherical tip (contact radius = 8 nm) that is used in Fig. 3, and the *lower panel* corresponds to a flat-ended tip (contact radius = 15 nm). Compared to the spherical one, the flat-ended tip usually imparts a weaker stress on the STO surface underneath. Since the downward $E_z^{dep}$ scales with the stress-gradient underneath the tip, the flat-ended tip induces a very small downward $E_z^{dep}$ (Fig. 4b, *upper panel*). In contrast, the lateral deformation (indicated by *curved arrows* in Fig. 4a), which controls the depolarisation field around the contact edge, is alike for both geometries. Consequently, the flat-ended tip induces a depolarisation field distribution around the contact edge (Fig. 4b) similar to that of the spherical tip in Fig. 3c, d. Additionally, an enhanced contact radius enlarges its spatial extent. A selective suppression of $E_z^{dep}$ underneath a flat-ended like tip should significantly reduce the surface-bulk $V_o^{..}$ migration. Meanwhile, the extended depolarisation field around the contact edge should improve the lateral transport of $V_o^{..}$.

To validate our proposition, we performed experiments with a sharp and blunt tip. The estimated radius of curvature of this blunt tip is larger than 200 nm (see Supplementary Fig. 7). Thus, it effectively yields a flat contact junction underneath the tip. We used the 120-uc thick STO film in these experiments. Figure 4c, d show the NVC maps after mechanical scans were performed with these tips at a contact force of 9.5 μN. Notably, we scanned both the left and right boundaries between the $V_o^{..}$-enriched and pristine regions.

To compare the sharp and blunt tip-induced $V_o^{..}$-redistribution we profiled the NVC maps, as indicated by vertical lines in Fig. 4c, d. Figure 4e, f show the corresponding NVC profiles. The overlapping NVC profiles along lines M1/M4 and M2/M3 demonstrate that the $V_o^{..}$-redistribution is reproducible for both tip geometries. However, the response of $V_o^{..}$ to the applied force from these two tips are clearly different. The NVC profiles in Fig. 4e exhibit a maximum drop in NVC of $\Delta_{max}^{dec} = -0.75$ along lines M2 and M3 but no appreciable increase in NVC ($\Delta_{max}^{inc}$) along lines M1 and M4. This implies that the sharp tip strongly depletes the $V_o^{..}$-enriched regions but barely enriches the pristine regions. This result is in qualitative agreement with that in Fig. 2e, which shows that the fraction of the depleted $V_o^{..}$ that laterally move with the tip progressively decreases for applied forces larger than 6 μN. The NVC profiles in Fig. 4f, however, exhibit a maximum drop in NVC of $\Delta_{max}^{dec} = -0.25$ along lines M2 and M3 and increase by $\Delta_{max}^{inc} = +0.2$ along lines M1 and M4. This implies that approximately 80% of the depleted $V_o^{..}$ laterally moved with the blunt tip. The strong reduction (improvement) of $\Delta_{max}^{dec}$ ($\Delta_{max}^{inc}$) thus confirms an active suppression of the out-of-plane migration and a simultaneous enhancement in the lateral transportation of $V_o^{..}$, during scans with the blunt tip. Overall, the ability to deterministically move $V_o^{..}$ constitutes the first experimental demonstration of a controlled manipulation of $V_o^{..}$ in an oxide and the resulting two-dimensional spatial modulation.

## Discussion

Through a combined experimental and theoretical approach, we demonstrated the flexoelectricity-mediated controlled manipulation of oxygen vacancies by the mechanical force from an SPM tip. A deterministic reconfiguration of spatial vacancy profile provides control over the electron density and related electronic correlation effects. This could enable, using an SPM-based all-in-one platform, the investigation of mesoscale quantum phenomena in oxides[43]. Ultimately, this creates the opportunity for developing mechanically sketched oxide devices, and ambipolar mechanical control of device functionalities such as electroresistance states. The voltage-free operation of the SPM tip would thereby eliminate the possibility of surface charging. At this point, we want to emphasise that flexoelectricity is a universal phenomenon, which can occur in any dielectric[18, 44]. However, the flexoelectric coefficients of few oxides, such as SrTiO₃ and BaTiO₃, are currently known[45]. Thus, our work should motivate the study of flexoelectricity in other oxides.

Broadly speaking, our KPFM-based imaging approach offers a time-efficient way of characterising the activation barrier potential for oxygen vacancy migration at room temperature to complement the conventional Arrhenius analysis[46]. Combined with the feasibility of determining the diffusion coefficient, this technique could thus become an essential metrology tool for oxide-based energy and memory research. Furthermore, our theoretical model that couples the phase-field simulations to the Nernst-Planck equation, can be employed to elucidate how depolarisation fields cause oxygen vacancies., electrons, and holes to redistribute. Therefore, the model can be extended to the study of emergent problems such as the domain wall conductivity, high electrical conductivity of morphotropic phase boundaries, and leakage current in ferroelectric oxides[47–50].

## Methods

**Thin film growth**. SrTiO₃ thin films were homoepitaxially grown on TiO₂-terminated Nb:SrTiO₃ (0.5% wt. doped) substrates using pulsed laser deposition technique. The growth dynamics and thickness were monitored by in-situ reflection high energy electron diffraction technique. The depositions were performed at 1000 °C and using an oxygen partial pressure of $5 \times 10^{-7}$ torr. After deposition, films were annealed at 800 °C for an hour in a 1 torr oxygen atmosphere and subsequently cooled down to room temperature at a cooling rate of 20 °C per minute.

**Kelvin probe force microscopy**. KPFM measurements were carried out using the Asylum Research Cypher SPM at room temperature and under ambient conditions. Pt/Ir-coated metallic tips (NANOSENSORS™ PPP-NCHPt) with a nominal spring constant ≈ 40 N/m were used for electrical/mechanical scans and KPFM imaging. The KPFM measurements were obtained in the non-contact mode using a lift height of 30 nm and the typical scan parameters used are as follows: $V_{ac} = 1$ V (peak-to-peak), $f_{resonance} = 250$ kHz, and scan rate = 1 Hz. Before each experiment, the spring constant of the cantilever was accurately determined from force–distance measurements and thermal tuning methods. The contact force during mechanical scans was varied accordingly by controlling the set-point voltage.

**Theoretical modelling**. To model the oxygen vacancy redistribution by mechanical force, we performed phase-field simulations by coupling the time-dependent Ginzburg-Landau and Nernst-Planck equations.

$$\frac{\partial \mathbf{P}}{\partial t} = -L \frac{\delta F}{\delta \mathbf{P}} \tag{2}$$

$$\frac{\partial [V_o^{..}]}{\partial t} = \nabla \left( D_{V_o^{..}} \nabla [V_o^{..}] + \mu_{V_o^{..}} [V_o^{..}] \nabla \phi \right) \tag{3}$$

In Eq. (2), $\mathbf{P}$ is the polarisation vector, $L$ is the kinetic coefficient and $F$ is the total free energy of the system, which includes Landau, electric, gradient and flexoelectric energy contributions. In Eq. (3), $[V_o^{..}]$, $D_{V_o^{..}}$, $\mu_{V_o^{..}}$, and $\phi$ denote the concentration, diffusion coefficient, and mobility of the oxygen vacancies and the electric potential, respectively. Detailed descriptions of the energy functional F and other relevant simulation parameters are presented in the Supplementary Note 4.

To solve Eqs. (2) and (3), the system was discretised into $100\Delta x \times 50\Delta y \times 500\Delta z$ grid points to implement the semi-implicit Fourier method[45]. The 120-uc-thick STO film is simulated to be $480\Delta z$ in thickness, while the substrate and air are each $10\Delta z$ in thickness. The parameters for the total free energy of STO were adopted from work of Y.L. Li et al.[46]. Following the work of R. Moos et al.[47], the initial concentration of oxygen vacancies in STO is assumed to be $3.66 \times 10^{14}$ cm$^{-3}$ based on our thin film growth conditions. In addition, the diffusion coefficient is assumed to be constant with regard to pressure and calculated as $1.23 \times 10^{-15}$ cm$^2$ s$^{-1}$ at room temperature[47]. A sufficiently long simulation time is used to ensure that the induced polarisation and oxygen vacancy concentration reach a quasi-steady state. At each time step, the electrostatic and elastostatic equilibrium equations are solved under the electric short-circuit[48] and mechanical mixed boundary conditions[49], respectively.

**Data availability**. The data that support the findings of this study are available from the corresponding authors upon reasonable request.

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

## Acknowledgements

This work was supported by the Institute for Basic Science in Korea (Grant No. IBS-R009-D1). B.W. and L.-Q.C acknowledges the support by the National Science Foundation under the grant number DMR-1410714 and by the Penn State MRSEC, Center for Nanoscale Science, under the award NSF DMR-1420620. Y.C. and S.V.K were supported by the U.S. DOE, Office of Basic Energy Sciences (BES), Materials Sciences and Engineering Division (MSED) under FWP Grant No. ERKCZ07 (Y.C., S.V.K.). A portion of this research was conducted at the Center for Nanophase Materials Sciences, which is a DOE Office of Science User Facility. We would like to thank Prof. Jong-Gul Yoon and Prof. Jin-Seok Chung for discussions. We also acknowledge Dr Luke Sandilands and John Henry Gruenewald for carefully proofreading the manuscript.

## Author contributions

S.D. conceived and planned this project under the direction of T.W.N. S.D. grew the films and performed structural characterisation with assistance from M.K. S.D. carried out KPFM measurements assisted by Y.J.S., S.M.Y., and L.F.W. and performed data analysis. B.W. and Y.C. performed theoretical modelling under the direction of L.-Q.C. and S.V.K. M.R.C. performed SEM measurements. S.D. and T.W.N. wrote the manuscript with inputs from all authors.

## Additional information

**Competing interests:** The authors declare no competing financial interests.

