## [Peer Review File · Nature Communications]

Reviewers' comments:

Reviewer #1 (Remarks to the Author):

The paper by Saikat Das reports on the manipulation of oxygen vacancies (VO). For that the authors investigate the effect of the force applied with a scanning probe microscope; while detecting the VO content by means of Kelvin probe force microscopy (KPFM). By combining the experiments with phase-field simulations, they argue that the redistribution of VO is caused by the depolarisation field associated with the stress-gradient. It is shown that the VOs migrate away from the tip, towards the bulk both in depth and laterally. However this can be controlled to certain extent by changing the geometry of the tip: flatter tips give rise to significant more lateral migration.

The paper reports on a problem of the highest current interest and importance, contains beautiful and meaningful experiments with convincing analysis and it is clearly written. In my opinion, it can be published in Nat. Comm. if the authors address the few points:

- 1) The first point is general: VO would also migrate driven solely by strain gradients (as VO are more stable in larger unit cells) even if no flexoelectric polarization was induced. It would help if the paper would clearly address the relative magnitude of these two effects.
- 2) It is mentioned that the technique used is "Kelvin probe force microscopy (KPFM) based novel imaging scheme". However, the technique itself does not seem novel. Please clarify.
- 3) The results section starts with a paragraph mentioning: "As shown in Fig. 1b (upper panel), the accumulation of VO at the STO surface locally alters this CPD. This change ...vacancy concentration."

This paragraph is very confusing because the authors have not explained yet how the VO-rich region has been created. Only in the next paragraph it is mentioned that this is done by poling.

- 4) "An identical functional dependence of NVC on the applied bias and force implies that the force functionally acts as a positive bias along M"

In my opinion, it cannot be convincingly proven that the data cannot be fitted by another functional as well. It would be more correct to say that positive bias and force have the same qualitative effect.

- 5) A suitable reference should be mentioned after:

"The stress-gradient from the SPM tip locally polarises STO through the flexoelectric effect"

- 6) I believe the paper should discuss tips instead of indenters

7) "This could enable, using an SPM-based all-in-one platform, the investigation of quantum phase transitions and emergent phenomena in mesoscopic limits". Please explain "in mesoscopic limits"

- 8) "Therefore, the model can be extended to the study of exotic conducting properties of charged domain walls and morphotropic phase boundaries in ferroelectric oxides^{42,43,44} "

No reference to morphotropic phase boundaries is given and it is not clear to me how MPB's are relevant to this problem.

Reviewer #2 (Remarks to the Author):

The paper deals with an approach on controlling distribution of oxygen vacancies in SrTiO₃ using scanning probe microscopy. The topic itself is important for a number of applications and it is very worth to study. However, the paper shows a few weak points which are addressed below.

1. The section describing KPFM measurements does not show much new data. It is to a larger extent a repetition of study published in ref. 27. Moreover, the authors do not consider a “quasi character” of oxygen vacancies. They only discuss the diffusion but do not take into account a possibility of recombination with oxygen which is very possible effect at the surface. What is the mechanism of only vertical diffusion of the vacancies? How deep they are able to diffuse and what is the reason for difference between 14- and 120-uc thick STO layers when imaged by KPFM? Kelvin probe is a surface technique which obviously does not allow probing as deep as 100 uc.

2. In the next section, the movement of vacancies by applied bias is discussed. However, when creating NVC map the role of tip in polarization of STO during KPFM scans is not taken into account. As shown in D.A. Nielsen et al. J. Appl. Phys. 118 (2015) 195301, the bias applied to the tip leads to polarization of a dielectric that significantly affects mapping of the surface potential. As STO is ferroelectric material this phenomenon must be even stronger.

3. In the part describing movement of vacancies by a contact force, the agreement between the model and measurements is very qualitative. What is concentration of vacancies that are moved? Is it realistic to move them by one tip pass? Is it possible to move them without a recombination over micron-range distances? These points are not addressed.

Summarising, the paper presents interesting approach on a possibility of redistribution of vacancies by SPM in STO. However, the suggested model for this phenomenon is weakly proved. The paper is not recommended for publication in Nature Communications.

Reviewer #3 (Remarks to the Author):

This manuscript describes a method to rearrange in a predictive way oxygen vacancies (OV) in STO using mechanical force from a scanning probe microscope (SPM). Since OV affect many properties of oxides, this work opens new scientific and technological avenues. Manipulating OV through the bias or force applied through a scanning probe is not a new idea. The main point of the paper is that, by understanding the mechanisms underlying OV migration under force, this manipulation can be performed in a controlled manner. The idea is interesting and the manuscript contains many interesting experimental observations and theoretical modelling. It is well written as well. However, some of the key conclusions are not fully supported by the presented data, and therefore it should be revised prior to publication.

1. A crucial claim is that KPFM provides a measurement of OVs. This is substantiated by an argument relying on thickness-dependent diffusion time for a charge carrier diffusing in the bulk. The data in figure 1 is interesting but not completely explained. For instance, why is the intensity of the signal in 1c so much weaker than in 1d at later time? With the simple 1D diffusion model, which seems appropriate given the dimensions of the perturbed region, much more information could be obtained from the data. Also, according to the definition in Eq. S1, the curves in Fig 1e should tend to 1. Why don't they?

2. The data presented in Fig 2 is very important, in that it demonstrates a distinct way of manipulating OV by force (as compared to by voltage). The authors show that under mechanical force, the region adjacent to a OV-rich region is enriched with OVs. They also show that this increase in OVs is much smaller than the decrease in the OV-rich region. They interpret this data as combination of downward bulk migration and partial lateral movement of OVs by the probe. The data is intriguing but information about the scanning protocol (in particular the fast scanning direction) is missing.

3. To understand the mechanical redistribution of OVs the authors present interesting theoretical modeling. They should emphasize that with the chosen parameters in the ferroelectric model, the material is in a paraelectric phase and therefore the induced polarization is a purely flexoelectric effect.

4. Based on this modeling, they put forth a mechanism, by which charged OV's would diffuse, either downward or sideways, as a result of the induced polarization. Incidentally, the authors qualify as "contentious" the conceptual framework behind this interpretation in the introduction. According to the modeling, a flat larger indenter would produce a smaller field in the vertical direction, but a similar lateral field over a larger perimeter. Therefore, such an indenter would produce weaker bulk diffusion but larger lateral migration. This is experimentally tested in Figure 4, where a weaker decrease of KPFM signal is observed in the OV-rich region and a larger increase is observed outside of this region as a result of scanning under force. The weakest point in this study is that the quantification of bulk vs lateral OV migration as a function of the tip geometry is not done in the same sample, or at least experimental setup. It is difficult to compare Figs 4d and 2d. These figures are very difficult to compare, and therefore do not provide compelling evidence of the predictive manipulation of OV's.

5. Furthermore, to support their proposed mechanism, which is in their words contentious, the authors could provide a number of controlled experiments. For instance, they could scan by applying force only during trace (and not during retrace). This could rule out a simple mechanical dragging of charged adsorbates during scanning if the results were insensitive to whether force was applied from left to right or from right to left (according to their proposed mechanism this should not matter). Furthermore, if diffusion was the transport mechanism, then the results should be sensitive to scanning velocity. Finally, their proposed mechanism should produce a smaller OV enrichment the farther apart from the OV-rich region.

Reviewer #1 (Remarks to the Author):

The paper by Saikat Das reports on the manipulation of oxygen vacancies (VO). For that the authors investigate the effect of the force applied with a scanning probe microscope; while detecting the VO content by means of Kelvin probe force microscopy (KPFM). By combining the experiments with phase-field simulations, they argue that the redistribution of VO is caused by the depolarisation field associated with the stress-gradient. It is shown that the VOs migrate away from the tip, towards the bulks both in depth and laterally. However this can be controlled to certain extent by changing the geometry of the tip: flatter tips give rise to significant more lateral migration.

The paper reports on a problem of the highest current interest and importance, contains beautiful and meaningful experiments with convincing analysis and it is clearly written. In my opinion, it can be published in Nat. Comm. if the authors address the few points:

Q.1a

The first point is general: VO would also migrate driven solely by strain gradients (as VO are more stable in larger unit cells) even if no flexoelectric polarization was induced. It would help if the paper would clearly address the relative magnitude of these two effects.

A. 1a

We would like to thank the reviewer for highlighting this important point that we did not address in the original manuscript. We agree that in the absence of flexoelectric polarization, VO could also migrate under the SPM tip-induced stress/strain through a mechanism known as the converse Vegard effect [PRB 83, 195313 (2011)]. This effect is characterized by the so-called Vegard coefficient $\beta_{ij} = \frac{\partial \varepsilon_{ij}}{\partial N_d}$, which relates the concentration of VO, N_d to the strain ε_{ij} . Therefore, it is worthwhile comparing the relative magnitude of the flexoelectric and the converse Vegard effect. However, for such comparison, it is important properly choosing the Vegard and flexoelectric coefficients. The flexoelectric coefficients of STO have been evaluated experimentally as well as theoretically, and they are comparable in order of magnitude. Unfortunately, the Vegard coefficients of STO are only computed theoretically [PRB 80, 064108 (2009)]. Therefore, without verifying these theoretical Vegard coefficients with experiments, a meaningful comparison between the flexoelectric and converse Vegard effect is difficult.

Nonetheless, A. N. Morzokova *et.al.*, have theoretically explored the impact of the flexoelectric and the converse Vegard effect under an applied force from an SPM tip in functional properties of PbTiO₃ (PTO) films [PRB 94, 174101 (2016)]. Regarding carrier (electrons) modulation, they found that the flexoelectric effect overpowers the converse Vegard effect. The carrier concentration in PTO is directly linked to the concentration of donors, which they assumed to be either VO or impurity ions. Notably, they used the Vegard coefficient of STO, and the flexoelectric coefficients in their calculation are comparable with ours. In light of these observations, we thus believe that in STO, in particular, the flexoelectric effect dominates over the converse Vegard effect.

To comply with the reviewer's suggestion, we have included the following discussion in line #179 of the revised manuscript.

“To understand the mechanical redistribution of V_o we first considered two mechanisms—the converse Vegard effect and the flexoelectric effect^{15,16}. Recently, the magnitude of these two effects under an applied force from SPM tip has been compared in a PbTiO₃ (PTO) thin film³⁸. This study suggests that the converse Vegard effect is much weaker than the flexoelectric effect. Notably, both the PTO and STO have comparable flexoelectric and Vegard coefficients^{25,38,39}, which determine the relative contributions of these two effects for a given force. Based on these considerations, we thus conclude that the

flexoelectric effect predominantly causes the mechanical redistribution of V_o , and the contribution from the converse Vegard effect is marginal.

Q.1b

It is mentioned that the technique used is "Kelvin probe force microscopy (KPFM) based novel imaging scheme". However, the technique itself does not seem novel. Please clarify.

A.1b

We agree with the reviewer's comment that the word "novel" is unnecessary. Therefore, we have now removed this "novel" word from the sentence in line # 62 of the revised manuscript.

Q.1c

The results section starts with a paragraph mentioning: "As shown in Fig. 1b (upper panel), the accumulation of VO at the STO surface locally alters this CPD. This change ...vacancy concentration."

This paragraph is very confusing because the authors have not explained yet how the VO-rich region has been created. Only in the next paragraph it is mentioned that this is done by poling.

A.1c

Acknowledging the reviewer's concern, we have replaced the original sentence with the following one in line # 76 of the revised manuscript.

"However, as schematically illustrated in Fig. 1b (upper panel), this CPD would change if the STO surface contains V_o , which can be locally accumulated by an electrical poling."

Q.1d

"An identical functional dependence of NVC on the applied bias and force implies that the force functionally acts as a positive bias along M"

In my opinion, it cannot be convincingly proven that the data cannot be fitted by another functional as well. It would be more correct to say that positive bias and force have the same qualitative effect.

A.1d

We agree with the reviewer's comment. We have now replaced the sentence with the following one in line #146 of the revised manuscript.

"This functional analysis suggests that the applied positive bias and force have a same qualitative effect along the lines E and M, respectively."

Q.1e

A suitable reference should be mentioned after:

"The stress-gradient from the SPM tip locally polarises STO through the flexoelectric effect"

A.1e

It is well established that a single crystalline STO slab, when subject to an inhomogeneous strain, develops a macroscopic polarisation due to the flexoelectric effect [PRL 99, 167601 (2007)]. When subject to the stress-gradient from an SPM tip, the STO thin film also develops a polarisation through this flexoelectric effect. We have therefore referred to this earlier work as the corresponding reference.

Q.1f

I believe the paper should discuss tips instead of indenters

A.1f

Following the reviewer's recommendation, we have replaced the word "indenter" with "*tip*" in the revised manuscript.

Q.1g

"This could enable, using an SPM-based all-in-one platform, the investigation of quantum phase transitions and emergent phenomena in mesoscopic limits". Please explain "in mesoscopic limits"

A. 1g

We acknowledge that the semantic "in mesoscopic limits" was misleading. Therefore, we have replaced the original sentence with the following one in line # 276 of the revised manuscript.

"This could enable, using an SPM-based all-in-one platform, the investigation of mesoscale quantum phenomena in oxides."

Q.1h

"Therefore, the model can be extended to the study of exotic conducting properties of charged domain walls and morphotropic phase boundaries in ferroelectric oxides^{42,43,44} "

No reference to morphotropic phase boundaries is given and it is not clear to me how MPB's are relevant to this problem.

A.1h

The theoretical model that we have developed in this work is not only limited to the study of VO-redistribution by applied bias/force. In general, it could provide a better understanding of some emergent problems in ferroelectric oxides. Examples include the domain wall conductivity [Nat. Mater. 14, 407 (2015)], leakage current [Acta Mater. 112, 230 (2016)], and high electrical conductivity of MPB's in BFO [Adv. Mater. 26, 4376 (2014)]. To clarify our message, we have replaced the original sentence with the following one in line #290 of the revised manuscript.

"Therefore, the model can be extended to the study of emergent problems such as the domain wall conductivity, high electrical conductivity of morphotropic phase boundaries, and leakage current in ferroelectric oxides^{47,48,49,50}."

Reviewer #2 (Remarks to the Author):

The paper deals with an approach on controlling distribution of oxygen vacancies in SrTiO₃ using scanning probe microscopy. The topic itself is important for a number of applications and it is very worth to study. However, the paper shows a few weak points which are addressed below.

Q. 2a

The section describing KPFM measurements does not show much new data. It is to a larger extent a repetition of study published in ref. 27 (Q.2a1). Moreover, the authors do not consider a “quasi character” of oxygen vacancies. They only discuss the diffusion but do not take into account a possibility of recombination with oxygen which is very possible effect at the surface (Q.2a2). What is the mechanism of only vertical diffusion of the vacancies (Q.2a3)? How deep they are able to diffuse and what is the reason for difference between 14- and 120-uc thick STO layers when imaged by KPFM (Q.2a4)? Kelvin probe is a surface technique which is obviously does not allow probing as deep as 100 uc.

A.2a

The reviewer has raised multiple questions. Thus, we have split our response into following subsections.

A.2a1

The section describing KPFM measurements on oxygen vacancy distribution elaborates following points.

- 1) The concept of detecting oxygen vacancies ($V_o^{\bullet\bullet}$) with the KPFM technique.
- 2) Proof of concept application of the KPFM technique to study the diffusion of $V_o^{\bullet\bullet}$.
- 3) The charges accumulated on the STO surface as a result of the electrical poling are $V_o^{\bullet\bullet}$ but not H^+/OH^- .
- 4) $V_o^{\bullet\bullet}$ are very stable on the surface of the 120-uc thick STO film.

The KPFM study published in ref. 27 [Nanoscale 7, 14351 (2015)], shows that the surface potential of a 3.2 nm thick STO film relaxes with time. This relaxation of the surface potential was attributed to be driven by the diffusion of $V_o^{\bullet\bullet}$ and/or by the surface reaction-mediated recombination of $V_o^{\bullet\bullet}$ with oxygen from the ambient. However, this work does not elucidate how the degree of equilibrium, $S(t)$ evolves with time and how does this time evolution vary with the film thickness. Therefore, we believe that our work contains new information, which provides a deeper insight into the time-dependent relaxation of surface potential in STO films.

We admit that gaining a better understanding of diffusion is not the main motivation of our work. However, elaborating points #1-3 is essential for discussing the main results of our manuscript. Furthermore, this diffusion study enabled us selecting the film with proper thickness (120-uc), which ensured a negligible contribution from the diffusion during the mechanical manipulation of $V_o^{\bullet\bullet}$.

A.2a2

We would like to thank the reviewer for raising the issue of recombination of $V_o^{\bullet\bullet}$ with oxygen at the surface, which we have missed discussing in the original manuscript. The recombination of $V_o^{\bullet\bullet}$ with oxygen from the ambient can occur through the so-called surface reaction process: $\frac{1}{2}O_2 + V_o^{\bullet\bullet} + 2e^- \rightleftharpoons O_o$. Here, O_o refers to oxygen on a regular lattice site [Sensors Actuators B Chem. 7, 763 (1992)]. While both the diffusion and surface reaction could explain the time-dependent decay of vacancy concentration at the surface, the dominating process can be readily identified by studying the time-dependency of the degree of equilibrium, $S(t)$. It has been extensively discussed in the literature [Sensors Actuators B Chem. 7, 763 (1992), Angew. Chem. Int. Ed. 47, 3874 (2008), and Nanoscale 7, 14351 (2015)] that if the surface reaction is the dominating process, the degree of equilibrium will exhibit a

linear time-dependency: $S(t) = \frac{k}{L}t$, where k is the surface reaction constant, and L is the sample thickness. However, if diffusion is the dominating process then $S(t)$ will exhibit a semi-parabolic time-dependency: $S(t) = \frac{4\sqrt{tD}}{L\sqrt{\pi}}$, where D is the diffusion coefficient. Our data (Fig. 1e of the revised manuscript) suggest that for both films, the time-dependency of $S(t)$ is semi-parabolic. Therefore, we argue that the surface reaction process is not dominating, and can be ignored.

A.2a3

We appreciate the reviewer's comment, $V_{\text{O}}^{\cdot\cdot}$ should diffuse both in the vertical and horizontal directions, which we have not addressed in the original manuscript. We have reexamined our KPFM images, and indeed the 120-uc thick STO film shows a signature of this horizontal diffusion along the STO surface. In Fig. R1 using KPFM images from Fig. 1d of the original manuscript we will elaborate this point. The KPFM profiles along the horizontal and vertical linecuts exhibit a broadening, which is marked by arrows in Figs. R1c and e. For quantifying this broadening, in Figs. R1d and f we plot the 1st derivative of these profiles. From a Gaussian fitting (not shown) we found that the full width at half maxima (FWHM) of these derivative profiles increases, in either direction, from about 0.5 to 0.8 μm .

The FWHM of the derivative profile is sensitive to the sharpness of boundary between the $V_{\text{O}}^{\cdot\cdot}$ -enriched and pristine regions, and the spatial resolution of KPFM measurement. Since the KPFM images in Figs. R1a and b were obtained using identical scanning parameters, and with the same SPM tip, the spatial resolutions of these two measurements should be identical. Hence, the increase of the FWHM suggests the sharpness of boundary between the $V_{\text{O}}^{\cdot\cdot}$ -enriched and pristine regions decrease with time—the $V_{\text{O}}^{\cdot\cdot}$ -enriched region laterally expands. This lateral expansion can be attributed to the diffusion of $V_{\text{O}}^{\cdot\cdot}$ in the horizontal direction, parallel to the STO surface.

The $V_{\text{O}}^{\cdot\cdot}$ -enriched region in the 14-uc thick STO film, however, does not show a discernible lateral expansion. The rapid relaxation of the KPFM contrast, inhibits us detecting the broadening of the KPFM profiles, even using the FWHM-analysis mentioned above.

Having demonstrated that $V_{\text{O}}^{\cdot\cdot}$ diffuse parallel to the STO surface, that is clearly discernible in the thicker STO film; we would like to remark that a meaningful extraction of D from this in-plane diffusion is not possible. The poor spatial resolution of the KPFM technique does not allow us to determine accurately how much the enriched region laterally expands.

We would also like to point out that while calculating $S(t)$ (see Supplementary Fig. 2 of the revised manuscript), we defined the concentration of $V_{\text{O}}^{\cdot\cdot}$ at any given time, t , as $[V_b(t) - V_{min}(t)]$. We used the pristine region and the center of the $V_{\text{O}}^{\cdot\cdot}$ -enriched region for extracting $V_b(t)$ and $V_{min}(t)$, respectively. Over time, a certain fraction of $V_{\text{O}}^{\cdot\cdot}$ those were initially located away from boundaries, including those at the center, would certainly diffuse in the lateral directions along the surface. On average this lateral diffusion, however, would not perturb the KPFM signal, and thus the $S(t)$. In contrast, owing to the high surface sensitivity of the KPFM technique, only the surface-bulk diffusion of $V_{\text{O}}^{\cdot\cdot}$ predominantly influences $S(t)$.

Figure R1 | **a-b**, KPFM images around a V_{O}^{\bullet} -enriched surface region of the 120-uc-thick STO film. Note that images **(a)** and **(b)** are the same as in Fig. 1d of the revised manuscript. **c-d**, Horizontal potential profiles **(c)** and their first derivatives **(d)**. **e-f**, Vertical potential profiles **(e)** and their first derivatives **(f)**. These potential profiles are taken from the KPFM images in **(a)** and **(b)**. The arrows in figures **(c)** and **(e)** mark the broadening due to the in-plane diffusion of V_{O}^{\bullet} . We have used Fig. **R1** as the Supplementary Fig. 3 of the revised manuscript.

A.2a4

We would like to thank the reviewer for pointing out that a proper discussion on the difference between the KPFM images of the 14-uc and 120-uc thick samples was missing in the original manuscript. We also agree with the reviewer that the KPFM technique is surface sensitive, and would not be able to probe how deep V_{O}^{\bullet} migrate in the vertical direction. However, a clear difference between these two samples is evident from Figs. 1c-e in the revised manuscript: the temporal evolution of the KPFM contrast and $S(t)$ are faster in the 14-uc thick STO film than in the 120-uc thick one. Through following discussions, we would like to argue that this difference complies with the diffusion characteristics of V_{O}^{\bullet} .

First, we note that the first step of this study was enriching the STO surface with V_{O}^{\bullet} , which was done by poling with a tip bias of -5V, while the bottom electrode was grounded. In Fig. R2 we compared the depth profiles of the simulated electrical potential inside the STO films under an applied tip bias. In this simulation we assumed that the the top electrode is biased: $V(x=0 \text{ nm}) = -5\text{V}$ and the bottom electrode is grounded: $V(x = \text{film thickness in nm}) = 0\text{V}$. Figure R2 shows that the electrical potential gradually decays over the film thickness. The tip bias applied during the poling perturbs the equilibrium distribution of V_{O}^{\bullet} in STO. Based on the simulated potential profiles in Fig. R2, we can therefore argue that this perturbation spans the entire film thickness. In this non-equilibrium state, vacancies in larger concentration would accumulate around the surface region. Upon removing the bias, the system would relax to its equilibrium state through the vacancy diffusion.

Figure R2 | Simulated electrical potential profiles inside the 14-uc and 120-uc thick STO films under an applied tip bias of -5V. This simulation was performed assuming that the top electrode is biased, while the bottom electrode is grounded, and the relative permittivity of STO is 10. We have used Fig. R2 as the Supplementary Fig. 4 of the revised manuscript.

Second, we note that the relaxation time for this diffusion-mediated equilibrium process should be proportional to the square of the film thickness [Sensors Actuators B Chem. 7, 763 (1992), Angew. Chem. Int. Ed. 47, 3874 (2008), and Nanoscale 7, 14351 (2015)].

Third, we note a salient feature of $V_o^{\bullet\bullet}$ diffusion, which is associated with the electrically charged nature of $V_o^{\bullet\bullet}$. Molecular dynamics simulations of $V_o^{\bullet\bullet}$ -diffusion in STO [J. Phys.: Condens. Matter 24, 485002 (2012)] suggest that during diffusion, the repulsive interaction between charged $V_o^{\bullet\bullet}$ inhibits them from moving independently. Note that this argument only holds for an ensemble of $V_o^{\bullet\bullet}$ but not for an isolated vacancy.

Based on above considerations we can argue that the applied bias during the poling perturbs a smaller volume in the 14-uc thick sample than in the 120-uc thick one. Thus, the thinner sample would relax faster than the thicker one. The KPFM technique only probes the surface layer, and the surface-bulk diffusion of $V_o^{\bullet\bullet}$ predominantly causes the KPFM contrast or $S(t)$ to change over time. However, owing to the correlated movement of $V_o^{\bullet\bullet}$ during the diffusion, the temporal evolution of the KPFM contrast and $S(t)$ reflect how the entire perturbed volume equilibrates after the poling. Connecting these logics thus clarifies the difference between the KPFM images of the 14-uc and 120-uc thick STO films.

The validity of above arguments is established by the excellent agreement between the diffusion coefficients D ($= 9.4(3) \times 10^{-19}$ and $3.8(2) \times 10^{-18}$ cm^2/s for the 14-uc and 120-uc thick STO film, respectively), which were obtained by fitting the time evolution of $S(t)$ with the Fick's 2nd law of diffusion.

For addressing questions Q. 2a2-Q. 2a4, we made following modifications.

1) To mention this surface reaction process, we have replaced the text “In contrast, because of diffusion, the vacancy concentration is expected to decrease with a strong thickness-dependent timescale³¹.” with “In contrast, because of diffusion or surface reaction that enables the recombination of $V_o^{\bullet\bullet}$ with oxygen from the ambient, the vacancy concentration is expected to decrease with a pronounced thickness-dependent timescale³².” in line # 91 of the revised manuscript. Also, we have replaced the text “Therefore, we argue that the surface charging is caused by $V_o^{\bullet\bullet}$, which undergo diffusion with time.” with “Therefore, we argue that the surface charging is caused by $V_o^{\bullet\bullet}$, which either undergo diffusion or recombine with oxygen from the ambient.” in line # 95 of the revised manuscript.

2) We have replaced the text “This diffusion process can be quantified in terms of the degree of equilibrium, $S(t)$, which describes the time evolution of the vacancy concentration (see Supplementary Sec. S1). Figure 1e plots $S(t)$ as a function of time. By fitting these data with Fick’s 2nd law of diffusion (solid lines), we obtained the diffusion coefficient $D = 9.4(3) \times 10^{-19}$ and $3.8(2) \times 10^{-18} \text{ cm}^2/\text{s}$ for the 14-uc and 120-uc-thick STO film, respectively. These values agree well with that of the bulk value $D_{\text{bulk}} \approx 10^{-(17\pm 3)} \text{ cm}^2/\text{s}$ (300K-extrapolated)³², which corroborates that the diminishing image contrast in Figs. 1c-d is consistent with the diffusion of V_{O}^{\bullet} . Furthermore, a reasonable agreement between D and D_{bulk} also validates the conceptual schematic depicted in the lower panel of Fig. 1b. Subsequently, we utilised this correlation between the vacancy concentration and KPFM signal to image the vacancy redistribution under an applied bias and force.” with the following in line # 97 of the revised manuscript.

“We can distinguish the dominating mechanism that causes the vacancy concentration to decay over time by analysing the time-dependency of the degree of equilibrium, $S(t)$ that can be calculated from the KPFM images (see Supplementary Sec. S1). The time-dependency of $S(t)$ describes how the surface equilibrates after the electrical poling. $S(t)$ will exhibit a semi-parabolic (linear) time-dependency if diffusion (surface reaction) is the dominating mechanism^{32,27}. Figure 1e plots $S(t)$ as a function of time. Evidently, the time-dependencies of $S(t)$ are semi-parabolic for both films—implying that the diffusion of V_{O}^{\bullet} causes the vacancy concentration to decrease over time.

Arguably, during the poling, the applied tip bias perturbs the equilibrium V_{O}^{\bullet} -distribution of the surface and the entire film underneath, which equilibrates through the diffusion of V_{O}^{\bullet} . This diffusion occurs both along the out-of-plane and in-plane directions. However, due to the high surface sensitivity of the KPFM technique, the surface-bulk diffusion of V_{O}^{\bullet} predominantly affects the time evolution of the KPFM contrast, and thus the time-dependency of $S(t)$. Notably, during this surface-bulk diffusion, the repulsive vacancy-vacancy interaction inhibit V_{O}^{\bullet} to migrate independently along the out-of-plane direction³³. Thus, the time-dependency of $S(t)$ effectively describes how the perturbed volume under the poled area equilibrates. Naturally, this volume would be smaller in the thinner STO film, and thus would equilibrate faster (see Supplementary Sec. S1 for a detailed discussion). This explains why the KPFM contrast ($S(t)$) diminishes (grows) more rapidly in the 14-uc thick STO than in the 120-uc thick STO film (Figs. 1c-e).

Following the rationale above, we fit the time evolution of $S(t)$ with Fick’s 2nd law of diffusion (solid lines in Fig. 1e). From this fitting, we obtained the diffusion coefficient $D = 9.4(3) \times 10^{-19}$ and $3.8(2) \times 10^{-18} \text{ cm}^2/\text{s}$ for the 14-uc and 120-uc-thick STO film, respectively. These values are well in the range of the bulk value $D_{\text{bulk}} \approx 10^{-(17\pm 3)} \text{ cm}^2/\text{s}$ (300K-extrapolated)³⁴, which validates the conceptual schematic depicted in the lower panel of Fig. 1b. Subsequently, we utilised this correlation between the vacancy concentration and KPFM signal to image the vacancy redistribution under an applied bias and force.”

3) We have completely revised the supplementary section S1. We have discussed, in details, the in-plane diffusion of V_{O}^{\bullet} , and how the surface sensitive KPFM technique allows us to characterise the surface-bulk diffusion in STO films of two different thicknesses. We have included Fig. R1 and Fig. R2 as the Supplementary Fig. 3 and Fig. 4 of the revised manuscript.

Q.2b

In the next section, the movement of vacancies by applied bias is discussed. However, when creating NVC map the role of tip in polarization of STO during KPFM scans is not taken into account. As shown in D.A. Nielsen et al. J. Appl. Phys. 118 (2015) 195301, the bias applied to the tip leads to polarization of a dielectric that significantly affects mapping of the surface potential. As STO is ferroelectric material this phenomenon must be even stronger.

A.2b

We would like to thank the reviewer for bringing to our attention this important paper describing the influence of the tip bias on KPFM measurement. Before discussing the role of tip bias in our KPFM data, we would like to point out that our STO film is not ferroelectric (see Supplementary section S3 of the revised manuscript). Also, we performed KPFM measurements in the non-contact mode, whereby the tip and the STO surface were separated by an air-gap of 30 nm.

For an applied tip bias $V (= V_{ac} + V_{dc})$ and tip radius R , ignoring the radial distribution, the magnitude of the tip bias-induced electric field as a function of the tip-surface separation (lift height) z can be approximated as $E(z) = 2RV/z^2$ [Annu. Rev. Mater. Sci. 28, 101 (1998)]. Therefore, by comparing KPFM scans performed at different lift heights, we can readily assess the “role of tip in polarization of STO during KPFM scans”. Following this logic, we took KPFM images of the pristine STO surface, and of the V_o^{\bullet} -enriched surface by varying the lift height between 10-50 nm. We used the 120-uc thick STO film in this study.

Figures R3a-b show that the KPFM signal of the pristine STO surface uniformly changes by about 70 mV upon decreasing z from 50 nm to 10 nm. This change is marginal, only 10% of the measured value ($\sim 1V$). Notably, during the KPFM imaging, the voltage (V_{dc}) that the feedback loop applies to the tip for nullifying the force between the tip and the STO surface should be equal to this measured KPFM signal ($\sim 1V$). A simple algebraic calculation suggests that the strength of the tip bias-induced electric field should be 25 times larger at $z = 10$ nm than at 50 nm: $E (z = 10 \text{ nm}) = 25E (z = 50 \text{ nm})$. Comparatively, however, the corresponding change in the measured KPFM signal is negligible.

Next, we consider the effect of the tip bias while imaging the V_o^{\bullet} -enriched STO surface (Figs. R3 c-d). In the pristine region, we observed a marginal (~ 100 mV) change in the measured KPFM signal upon decreasing z from 50 to 10 nm. On the contrary, the KPFM contrast across the V_o^{\bullet} -enriched region shows a relatively larger change by about 500 mV, which is about 30% of the KPFM contrast obtained at $z = 50$ nm. Once again, this change is much smaller than the expected increase in the strength of the electric field from the corresponding z -variation.

We could think of a simple explanation for the relatively larger effect of z -variation on the KPFM contrast. V_o^{\bullet} are electrically charged, and the electric field emanating from them would induce an electrostatic force between the tip and V_o^{\bullet} . The magnitude of this force would depend on z , and on the concentration of V_o^{\bullet} . The magnitude of this force, however, would be independent of the applied tip bias (V) [Nanotechnology 28, 025703 (2017)]. Thus, during KPFM imaging, the closer the tip approaches towards the V_o^{\bullet} -enriched region, the stronger it feels this V_o^{\bullet} -induced force. Accordingly, the feedback loop utilizes a larger V_{dc} for nullifying the force acting between the tip and the V_o^{\bullet} -enriched STO surface. These considerations readily corroborate the relatively larger effect of z -variation on the KPFM contrast.

To sum up, based on our experiments, we believe that the influence of the tip bias and the resulting polarization of STO is very small, and is uniform across an entire KPFM image. While creating an NVC map from a KPFM image, we defined the concentration of V_o^{\bullet} to be equal to the difference between the KPFM signals measured from the pristine and V_o^{\bullet} -enriched regions. This approach readily removes this

Figure R3 | **a**, KPFM image of the pristine STO surface. This image was obtained with a lift height of 10 nm. **b**, The KPFM profiles (along the line cut in **a**) from KPFM images that were obtained by varying the lift height between 10-50 nm. **c**, KPFM image of the V_o'' -enriched STO surface. This image was obtained with a lift height of 10 nm. **d**, The KPFM profiles (along the line cut in **c**) from KPFM images that were obtained by varying the lift height between 10-50 nm.

uniform contribution of the tip bias. Further, we would like mention that for analysing the diffusion data we defined V_o'' -concentration the same way as for constructing the NVC maps. If the influence of the tip bias on the pristine region were different than on the V_o'' -enriched regions, then our data would contain an uncompressed contribution from the tip bias. This uncompensated contribution would then cause a large error in estimating the diffusion coefficients (D). However, the calculated diffusion coefficients are well in the range the expected bulk value. These considerations also testify that our approach of defining the concentration of V_o'' readily removes the contribution of the tip bias-induced STO polarisation during KPFM scans.

Nonetheless, taking into consideration the possible role of the tip bias in a KPFM measurement, we cited the paper: D.A. Nielsen et al. J. Appl. Phys. 118 (2015) 195301, as a reference, and included the following sentence in line #79 of the revised manuscript.

“Moreover, the tip bias used during the KPFM measurement has been argued to influence the measured CPD²⁹.”

Q.2c

In the part describing movement of vacancies by a contact force, the agreement between the model and measurements is very qualitative (Q.2c1). What is concentration of vacancies that are moved (Q.2c2)? Is it realistic to move them by one tip pass (Q.2c3)? Is it possible to move them without a recombination over micron-range distances (Q.2c4)? These points are not addressed.

A. 2c

The reviewer has raised multiple questions. Thus, we have split our response into following subsections.

A.2c1

The Observation made by the reviewer that “the agreement between the model and measurements is very qualitative.” is true. There are few reasons that do not allow to establish a quantitative agreement between the model and measurements. First, the KPFM technique, in its present form, doesn't allow us to determine the vacancy concentration quantitatively. Second, the following basic assumptions involved in the model, which are different from the experiment.

- 1) We used a sufficiently long simulation time for obtaining a quasi-steady state solution of the coupled time-dependent Ginzburg-Landau and the Nernst-Planck equations.
- 2) In our simulation the tip is static.

Thus, the simulated NVC map in Fig. 3b of the revised manuscript represents the upper limit of the flexoelectric effect-mediated V_o^{\bullet} -redistribution around the tip-STO contact junction. In the experiment, the tip, however, is mobile. Therefore, the magnitude of V_o^{\bullet} -redistribution would depend on the scanning velocity—longer the tip spends in contact with the V_o^{\bullet} -enriched surface, the larger number of vacancies it would redistribute. This experimental timescale cannot be compared with the simulation timescale.

A.2c2

The KPFM technique, in its present form, does not allow us to determine the vacancy concentration quantitatively. We have addressed this drawback in line # 80 of the revised manuscript: “*The influence of each factor cannot be individually separated, which inhibits a quantitative determination of the vacancy concentration with the KPFM technique*”. To circumvent this problem, in our manuscript, we used the NVC map, which allows us to elaborate the relative change in the V_o^{\bullet} concentration as a result of the applied tip bias/force. Accordingly, we have also adopted the same approach while showing the simulated V_o^{\bullet} -redistribution in our manuscript.

A.2c3

Indeed it is possible to move V_o^{\bullet} with one tip pass. Figure. R4 demonstrates the feasibility of moving oxygen vacancies with a single trace. However, the enrichment of the pristine region caused by this single line scan is very weak. Note that the high background within the V_o^{\bullet} -enriched region, inhibits us detecting the lateral migration of V_o^{\bullet} when the tip moves from the pristine into V_o^{\bullet} -enriched region. We have included Fig. R4 in the Supplementary information of the revised manuscript as Fig. 9.

A.2c4

We acknowledge the reviewer's concern about the possibility of recombination during the lateral motion of V_o^{\bullet} , which we missed discussing in the text. As we discussed in **A.2a2**, the contribution of the surface reaction, which could facilitate the recombination of V_o^{\bullet} with oxygen from the ambient is marginal. Besides, the tip bias-induced amplification of this surface reaction process [Nature Chem. 3, 707 (2011)] can be ruled out because the mechanical scan was performed using a grounded tip. Hence, there is a good reason for us to argue that the possibility of recombination during the lateral motion of V_o^{\bullet} over micron-range distances is negligible. For clarifying this point, we have included the following sentence in line #165 of the revised manuscript.

“Also, the possibility of recombination with oxygen from the ambient during the lateral motion of V_o^{\bullet} can be ignored based on the following considerations. The influence of the surface reaction process, which could facilitate this recombination is negligible in our film. Furthermore, the use of a grounded tip during the mechanical scan rules out the bias-induced amplification of this surface reaction process³⁷.”

Figure R4 | **a-b**, NVC maps after mechanical scans were performed with a load of 6 μN across the left **(a)** right **(b)** boundaries between V_o -enriched and pristine regions. During these scans, the tip was traced following a predefined defined path, which is marked by small triangles. The start (end) of the trace is marked by the green (red) triangle. **c-d**, NVC profiles along lines V_1 and V_2 on the map in **a** **(c)** and in **b** **(d)**. The NVC profile along the line V_1 (V_2) shows a decrease (increase) in V_o concentration inside (outside) the V_o -enriched regions. For clarity, we used vertical arrows to mark the increase in vacancy concentration in the lower panel of **c** and **d**. We have used Fig. **R4** as the supplementary Fig. 9 of the revised manuscript.

Q.2d

Summarising, the paper presents interesting approach on a possibility of redistribution of vacancies by SPM in STO. However, the suggested model for this phenomenon is weakly proved. The paper is not recommended for publication in Nature Communications.

A.2d

We would like to thank the reviewer for pointing out multiple weak points of the original manuscript. We believe that addressing these shortcomings has significantly improved our manuscript. For providing more compelling evidence to support our proposed model, we have now performed a controlled experiment; whereby we applied forced only during the trace and varied the scanning velocity. Figure R5 shows the result from this experiment. Here, using a sharp tip, we traced five lines across the boundary between a V_o -enriched and pristine regions, and we used two different scanning velocities. Figures R5a and b show the corresponding NVC map and NVC profiles.

Figure R5 | **a**, The NVC map after mechanical scans were performed with two different scanning velocities (0.5 and 20 μm/s) across the boundary between a $V_o^{\bullet\bullet}$ -enriched and pristine regions. Scans were performed with a load of 6 μN. The force was applied only during the trace, and the tip was lift-off during the retrace. The horizontal arrow marks the trace direction. **b**, NVC profiles along lines V_1 and V_2 , showing vacancy depletion inside the $V_o^{\bullet\bullet}$ -enriched and enrichment outside this region. The corresponding scanning velocities are indicated on the plot. The profiles are averaged over a 0.5 μm wide window. Note the region scanned with the tip velocity of 20 μm/s shows a large increase in NVC at the end of the trace. This is an artefact, caused by a larger deformation of the STO surface during the withdrawal of the tip from the STO surface. We often made similar observations even for an engagement of the tip on the pristine region. For profiling the NVC map in (a), we have thus excluded the left and right ends of the scanned regions. We have used Fig. R5 as the Supplementary Fig. 11 of the revised manuscript.

Figure R5b elaborates two points.

- 1) Regardless of the scanning velocity, the net decrease in NVC inside the $V_o^{\bullet\bullet}$ -enriched region is larger than the net increase in NVC outside this region.
- 2) The slower scan causes a larger depletion (enrichment) inside (outside) the $V_o^{\bullet\bullet}$ -enriched region.

The first observation qualitatively agrees with that in Fig. 2 of the revised manuscript: scanning with a sharp tip yields a stronger (weaker) depletion (enrichment) inside (outside) the $V_o^{\bullet\bullet}$ -enriched region. We proposed that the depolarisation field beneath the sharp tip preferentially drives $V_o^{\bullet\bullet}$ into the bulk, which leads to the stronger drop in NVC inside the $V_o^{\bullet\bullet}$ -enriched region. In contrast, a smaller fraction of the depleted $V_o^{\bullet\bullet}$ gets trapped around the contact edge and move with the tip-yielding a relatively weaker gain in NVC outside the $V_o^{\bullet\bullet}$ -enriched region. Notably, the mechanical redistribution of $V_o^{\bullet\bullet}$ shown in Fig. 2 of the revised manuscript was obtained by raster scanning the tip, whereby the force was applied both during the trace and retrace. Therefore, the result in Fig. 2 should be understood as an average response of $V_o^{\bullet\bullet}$ to the force applied during the trace and retrace. However, the $V_o^{\bullet\bullet}$ -redistribution shown in Fig. R5 was obtained by applying force only during the trace, which allows us to avoid the influence of the retrace. Nonetheless, there is a qualitative agreement between Fig. R5 and Fig. 2, which validates our proposed model.

The second observation substantiates our arguments that for an applied force, the longer the tip spends in contact with the surface, the larger number of $V_o^{\bullet\bullet}$ it redistributes (see A.2c1). Therefore, we believe that this result is line with our suggested model for mechanical redistribution of $V_o^{\bullet\bullet}$.

We have included Fig. R5 as the Supplementary Fig. 11. Moreover, we replaced the text “The simulation results qualitatively explain the characteristics of the mechanical $V_o^{\bullet\bullet}$ -redistribution in Fig. 2. During a mechanical scan, a spatially extended and strong downward E_z^{dep} field acts over a larger

fraction of V_o underneath the tip, which results in a dominant surface-bulk migration. In contrast, a small fraction of V_o becomes effectively trapped within a shallow annular region around the contact edge by the upward E_z^{dep} and inward $E_{x,y}^{dep}$. These trapped vacancies can move laterally with the tip from the V_o -enriched region to the pristine region during the tip's lateral motion. Therefore, the contrasting roles of the depolarisation field underneath the tip and around the contact edge corroborate both the dominant surface-bulk migration and the lateral motion of V_o with the tip along the surface. In the following, we illustrate that this depolarisation can be tailored in favour of the lateral motion of V_o , which enables controllably manipulating the vacancy distribution.” with the following one in line #216 of the revised manuscript.

“The simulation results qualitatively explain the characteristics of the mechanical V_o -redistribution in Fig. 2. During a mechanical scan, a spatially extended and strong downward E_z^{dep} field acts over a larger fraction of V_o underneath the tip, which results in a dominant surface-bulk migration. In contrast, a small fraction of V_o becomes effectively trapped within a shallow annular region around the contact edge by the upward E_z^{dep} and inward $E_{x,y}^{dep}$. These trapped vacancies can move laterally with the tip from the V_o -enriched region to the pristine region during the tip's lateral motion. Therefore, the contrasting roles of the depolarisation field underneath the tip and around the contact edge corroborate both the dominant surface-bulk migration and the relatively weaker lateral motion of V_o with the tip along the surface.

Notably, while scanning, the tip redistributes V_o regardless whether it moves from left-right or right-left. Thus, the mechanical redistribution of V_o should be understood as an average response of V_o to the force applied during the trace and retrace. Moreover, the scanning velocity would influence the V_o -redistribution—longer the tip spends in contact with STO, the larger number of V_o it would redistribute. To check whether these factors could contribute to the weaker lateral motion of V_o , we performed additional experiments, whereby we applied force only during the trace, and varied the scanning velocity (see Supplementary information). These experiments also yielded a weaker lateral motion but a dominant surface-bulk migration of V_o —highlighting the dominating influence of depolarisation field underneath the tip. In the following, we illustrate that the depolarisation field around the tip-STO contact junction can be tailored in favour of the lateral motion of V_o , which enables controllably manipulating the vacancy distribution.”

After addressing the weaker points mentioned by the reviewer, and incorporating new supportive evidence for our suggested model, we are optimistic that the revised manuscript should now be suitable for publication in *Nature Communications*.

Reviewer #3 (Remarks to the Author):

This manuscript describes a method to rearrange in a predictive way oxygen vacancies (OV) in STO using mechanical force from a scanning probe microscope (SPM). Since OV affect many properties of oxides, this work opens new scientific and technological avenues. Manipulating OV through the bias or force applied through a scanning probe is not a new idea. The main point of the paper is that, by understanding the mechanisms underlying OV migration under force, this manipulation can be performed in a controlled manner. The idea is interesting and the manuscript contains many interesting experimental observations and theoretical modelling. It is well written as well. However, some of the key conclusions are not fully supported by the presented data, and therefore it should be revised prior to publication.

Q. 3a

A crucial claim is that KPFM provides a measurement of OVs. This is substantiated by an argument relying on thickness-dependent diffusion time for a charge carrier diffusing in the bulk. The data in figure 1 is interesting but not completely explained. For instance, why is the intensity of the signal in 1c so much weaker than in 1d at later time (Q. 3a1)? With the simple 1D diffusion model, which seems appropriate given the dimensions of the perturbed region, much more information could be obtained from the data. Also, according to the definition in Eq. S1, the curves in Fig 1e should tend to 1. Why don't they (Q. 3a2)?

A. 3a

The reviewer has raised two questions. Thus, we have split our response into two following subsections.

A. 3a1

We would like to thank the reviewer for pointing out that there are many hidden aspects of the simple 1D model, which we had not considered. From Figs 1c and 1d of the revised manuscript, it is evident that the relaxation of the KPFM contrast is faster in the 14-uc thick STO film than in the 120-uc thick STO. Also, Fig. 1e shows the corresponding degree of equilibrium, $S(t)$ of the 14-uc thick film grows faster than that of the 120-uc thick one. Through following discussions, we would like to argue that these observations comply with the diffusion characteristics of OVs

First, we note that the first step of this study was enriching the STO surface with V_{O}^{\bullet} , which was done by poling with a tip bias of -5V, while the bottom electrode was grounded. In Fig. R1 we compared the depth profiles of the simulated electrical potential inside the STO films under an applied tip bias. In this simulation we assumed that the top electrode is biased: $V(x=0 \text{ nm}) = -5\text{V}$ and the bottom electrode is grounded: $V(x = \text{film thickness in nm}) = 0\text{V}$. Figure R1 shows that the electrical potential gradually decays over the film thickness. The tip bias applied during the poling perturbs the equilibrium distribution of V_{O}^{\bullet} in STO. Based on the simulated potential profiles in Fig. R1, we can therefore argue that this perturbation spans the entire film thickness. In this non-equilibrium state, vacancies in larger concentration would accumulate around the surface region. Upon removing the bias, the system would relax to its equilibrium state through the vacancy diffusion.

Second, we note that the relaxation time for this diffusion-mediated equilibrium process should be proportional to the square of the film thickness [Sensors Actuators B Chem. 7, 763 (1992), Angew. Chem. Int. Ed. 47, 3874 (2008), and Nanoscale 7, 14351 (2015)].

Third, we note a salient feature of V_{O}^{\bullet} diffusion, which is associated with the electrically charged nature of V_{O}^{\bullet} . Molecular dynamics simulations of V_{O}^{\bullet} -diffusion in STO [J. Phys.: Condens. Matter 24, 485002 (2012)] suggest that during diffusion, the repulsive interaction between charged V_{O}^{\bullet} inhibits them from moving independently. Note that this argument only holds for an ensemble of V_{O}^{\bullet} but not for an isolated vacancy.

Figure R1 | Simulated electrical potential profiles inside the 14-uc and 120-uc thick STO films under an applied tip bias of -5V. This simulation was performed assuming that the top electrode is biased, while the bottom electrode is grounded, and the relative permittivity of STO is 10. We have used Fig. **R1** as the Supplementary Fig. 4 of the revised manuscript.

Based on above considerations we can argue that the applied bias during the poling perturbs a smaller volume in the 14-uc thick sample than in the 120-uc thick one. Thus, the thinner sample would relax faster than the thicker one. The KPFM technique only probes the surface layer, and the surface-bulk diffusion of V_o^{\bullet} predominantly causes the KPFM contrast or $S(t)$ to change over time. However, owing to the correlated movement of V_o^{\bullet} during the diffusion, the temporal evolution of the KPFM contrast and $S(t)$ reflect how the entire perturbed volume equilibrates after the poling. Connecting these logics thus clarifies why the KPFM contrast in the 14-uc thick STO decays much faster compared to the 120-uc thick STO film.

The validity of above arguments is established by the excellent agreement between the diffusion coefficients D ($= 9.4(3) \times 10^{-19}$ and $3.8(2) \times 10^{-18}$ cm^2/s for the 14-uc and 120-uc thick STO film, respectively), which were obtained by fitting the time evolution of $S(t)$ with the Fick's 2nd law of diffusion.

For explaining the difference between the 14-uc thick and 120-uc thick STO-films, and addressing other relevant issues we made following modifications.

1) We have replaced the text “In contrast, because of diffusion, the vacancy concentration is expected to decrease with a strong thickness-dependent timescale³¹.” with “*In contrast, because of diffusion or surface reaction that enables the recombination of V_o^{\bullet} with oxygen from the ambient, the vacancy concentration is expected to decrease with a pronounced thickness-dependent timescale³².*” in line #91 of the revised manuscript. Also, we have replaced the text “Therefore, we argue that the surface charging is caused by V_o^{\bullet} , which undergo diffusion with time.” with “*Therefore, we argue that the surface charging is caused by V_o^{\bullet} , which either undergo diffusion or recombine with oxygen from the ambient.*” in line #95 of the revised manuscript.

2) We have replaced the text “This diffusion process can be quantified in terms of the degree of equilibrium, $S(t)$, which describes the time evolution of the vacancy concentration (see Supplementary Sec. S1). Figure 1e plots $S(t)$ as a function of time. By fitting these data with Fick's 2nd law of diffusion (solid lines), we obtained the diffusion coefficient $D = 9.4(3) \times 10^{-19}$ and $3.8(2) \times 10^{-18}$ cm^2/s for the 14-uc and 120-uc-thick STO film, respectively. These values agree well with that of the bulk value $D_{bulk} \approx 10^{-(17\pm 3)}$ cm^2/s (300K-extrapolated)³², which corroborates that the diminishing image contrast in Figs. 1c-d is consistent with the diffusion of V_o^{\bullet} . Furthermore, a reasonable agreement between D and D_{bulk} also validates the conceptual schematic depicted in the lower panel of Fig. 1b. Subsequently, we utilised

this correlation between the vacancy concentration and KPFM signal to image the vacancy redistribution under an applied bias and force.” with the following in line # 97 of the revised manuscript.

“We can distinguish the dominating mechanism that causes the vacancy concentration to decay over time by analysing the time-dependency of the degree of equilibrium, $S(t)$ that can be calculated from the KPFM images (see Supplementary Sec. S1). The time-dependency of $S(t)$ describes how the surface equilibrates after the electrical poling. $S(t)$ will exhibit a semi-parabolic (linear) time-dependency if diffusion (surface reaction) is the dominating mechanism^{32,27}. Figure 1e plots $S(t)$ as a function of time. Evidently, the time-dependencies of $S(t)$ are semi-parabolic for both films—implying that the diffusion of V_o causes the vacancy concentration to decrease over time.

Arguably, during the poling, the applied tip bias perturbs the equilibrium V_o -distribution of the surface and the entire film underneath, which equilibrates through the diffusion of V_o . This diffusion occurs both along the out-of-plane and in-plane directions. However, due to the high surface sensitivity of the KPFM technique, the surface-bulk diffusion of V_o predominantly affects the time evolution of the KPFM contrast, and thus the time-dependency of $S(t)$. Notably, during this surface-bulk diffusion, the repulsive vacancy-vacancy interaction inhibits V_o to migrate independently along the out-of-plane direction³³. Thus, the time-dependency of $S(t)$ effectively describes how the perturbed volume under the poled area equilibrates. Naturally, this volume would be smaller in the thinner STO film, and thus would equilibrate faster (see Supplementary Sec. S1 for a detailed discussion). This explains why the KPFM contrast ($S(t)$) diminishes (grows) more rapidly in the 14-uc thick STO than in the 120-uc thick STO film (Figs. 1c-e).

Following the rationale above, we fit the time evolution of $S(t)$ with Fick’s 2nd law of diffusion (solid lines in Fig. 1e). From this fitting, we obtained the diffusion coefficient $D = 9.4(3) \times 10^{-19}$ and $3.8(2) \times 10^{-18} \text{ cm}^2/\text{s}$ for the 14-uc and 120-uc-thick STO film, respectively. These values are well in the range of the bulk value $D_{\text{bulk}} \approx 10^{-(17\pm3)} \text{ cm}^2/\text{s}$ (300K-extrapolated)³⁴, which validates the conceptual schematic depicted in the lower panel of Fig. 1b. Subsequently, we utilised this correlation between the vacancy concentration and KPFM signal to image the vacancy redistribution under an applied bias and force.”

3) We have completely revised the Supplementary section S1. We have discussed, in details, how the surface sensitive KPFM technique allows us to characterise the surface-bulk diffusion of OV’s in STO films of two different thicknesses. We included Fig. R1 as the Supplementary Fig. 4 of the revised manuscript.

A. 3a2

Reviewer’s observation is correct that “according to the definition in Eq. S1, the curves in Fig 1e should tend to 1”. However, in Fig. 1e they do not tend to 1. An unintentional mistake caused this discrepancy. In the Supplementary information of the original manuscript, we referred to t_∞ as the time lag between the OV-enrichment process and the time the last KPFM image was acquired. However, t_∞ should be referred to as the time when the KPFM contrast disappears, implying, $[V_b(t_\infty) - V_{\text{min}}(t_\infty)] = 0$. We have corrected this error in the Supplementary information of the revised manuscript.

Q.3b

The data presented in Fig 2 is very important, in that it demonstrates a distinct way of manipulating OV by force (as compared to by voltage). The authors show that under mechanical force, the region adjacent to a OV-rich region is enriched with OVs. They also show that this increase in OVs is much smaller than the decrease in the OV-rich region. They interpret this data as combination of downward bulk migration and partial lateral movement of OVs by the probe. The data is intriguing but information about the scanning protocol (in particular the fast scanning direction) is missing.

A.3b

Following the reviewer's comment, we have marked the fast scanning direction by a double-headed arrow in Fig. 2 and Fig. 4 of the revised manuscript.

Q.3c

To understand the mechanical redistribution of OVs the authors present interesting theoretical modeling. They should emphasize that with the chosen parameters in the ferroelectric model, the material is in a paraelectric phase and therefore the induced polarization is a purely flexoelectric effect.

A. 3c

Following the reviewer's suggestion, we added the following sentence in line #198 of the revised manuscript.

"Since in the simulation the STO film is assumed to be in the paraelectric phase, the stress-gradient-induced polarisation in Fig. 3a purely stems from the flexoelectric effect."

Q.3d

Based on this modeling, they put forth a mechanism, by which charged OVs would diffuse, either downward or sideways, as a result of the induced polarization. Incidentally, the authors qualify as "contentious" the conceptual framework behind this interpretation in the introduction. According to the modeling, a flat larger indenter would produce a smaller field in the vertical direction, but a similar lateral field over a larger perimeter. Therefore, such an indenter would produce weaker bulk diffusion but larger lateral migration. This is experimentally tested in Figure 4, where a weaker decrease of KPFM signal is observed in the OV-rich region and a larger increase is observed outside of this region as a result of scanning under force. The weakest point in this study is that the quantification of bulk vs lateral OV migration as a function of the tip geometry is not done in the same sample, or at least experimental setup. It is difficult to compare Figs 4d and 2d. These figures are very difficult to compare, and therefore do not provide compelling evidence of the predictive manipulation of OVs.

A.3d

The observation made by the reviewer is partly correct. Figures 2d and 4d of the original manuscript were obtained from the same 120-uc thick STO film, and with the same experimental setup. However, the scanning parameters, namely the scanning velocity and the number of scan lines used in these two studies were not identical. Thus, we agree that ideally Fig. 2d and Fig. 4d cannot be compared. To resolve this issue, we performed an experiment with a sharp tip and the scanning parameters that were used for the scan with the blunt tip. Figure R2 shows the comparison between the sharp tip and blunt tip-induced mechanical redistribution of OVs. Figure R2b and R2d are the same as Figs. 4c and d of the original manuscript. Note that the KPFM image, which we used for constructing the NVC map in R2b was obtained using the blunt tip that was used for the mechanical scan. Changing tips within a reasonable time frame during the vacancy enrichment-mechanical scan-imaging sequence is not feasible. Because of the use of blunt tip, the boundary between the OV-rich and pristine regions appears more diffused in Fig. R2b than in Fig. R2a.

Figure R2 | **a-b**, NVC maps after mechanical scans were performed across the boundaries between vacancy-rich and pristine regions with the sharp tip (**a**) and blunt tip (**b**). Mechanical scans were performed with a contact force of about 9.5 μN and using identical scanning parameters. **c-d** NVC profiles along the vertical lines in **a** (**c**) and in **b** (**d**). Note that Figs. **R2b** and **R2d** are the same as Figs. 4c-d of the original manuscript. Figures **R2a-d** are now used as Figs. 4c-f of the revised manuscript.

Nonetheless, NVC maps in Fig. R2a and R2b illustrate a clear difference between the OV-redistribution caused by these two tips. The sharp tip causes a larger depletion inside the OV-rich region and a negligible enrichment outside of this region. Comparatively, the blunt tip causes a smaller depletion inside the OV-rich region but a larger enrichment in the region outside. NVC profiles in Figs. R2c and R2d quantitatively elaborate these points. Thus, we believe that this new result now provides more compelling evidence of the predictive manipulation of OVs. We have included Fig. R2 as Figs. 4c-f of the revised manuscript.

To specify that the comparison between the sharp and blunt tips is carried out using the same sample, we added the following text in line # 252 of the revised manuscript.

“We used the 120-uc thick STO film in these experiments.”

Furthermore, we replaced the text “To characterise this blunt tip-induced V_{O}^{\bullet} -redistribution we profiled the NVC map, as indicated by vertical lines in Fig. 4c. The corresponding NVC profiles are shown in Fig. 4d. The overlapping NVC profiles along lines M1/M4 demonstrate that the V_{O}^{\bullet} -redistribution is reproducible. The NVC profiles exhibit a maximum drop in NVC of $\Delta_{\text{max}}^{\text{dec}} = -0.25$ along lines M2 and M3 and increase by $\Delta_{\text{max}}^{\text{inc}} = +0.2$ along lines M1 and M4. This implies that approximately 80% of the depleted V_{O}^{\bullet} laterally moved with the tip. Furthermore, pristine regions across borders are twice as much populous as that in Fig. 2b. These observations confirm an active suppression of the out-of-plane migration and a simultaneous enhancement in the lateral transportation of V_{O}^{\bullet} , as we proposed based on the above simulation. Overall, the ability to deterministically move V_{O}^{\bullet} constitutes the first experimental demonstration of a controlled manipulation of V_{O}^{\bullet} in an oxide and the resulting two-dimensional spatial modulation.” with the following one in line # 256 of the revised manuscript.

“To compare the sharp and blunt tip-induced V_{O}^{\bullet} -redistribution we profiled the NVC maps, as indicated by vertical lines in Figs. 4c-d. Figures 4e-f show the corresponding NVC profiles. The overlapping NVC profiles along lines M1/M4 and M2/M3 demonstrate that the V_{O}^{\bullet} -redistribution is reproducible for both

tip geometries. However, the response of V_o to the applied force from these two tips are clearly different. The NVC profiles in Fig. 4e exhibit a maximum drop in NVC of $\Delta_{max}^{dec} = -0.75$ along lines M2 and M3 but no appreciable increase in NVC (Δ_{max}^{inc}) along lines M1 and M4. This implies that the sharp tip strongly depletes the V_o -enriched regions but barely enriches the pristine regions. This result is in qualitative agreement with that in Fig. 2e, which shows that the fraction of the depleted V_o that laterally move with the tip progressively decreases for applied forces larger than $6\mu N$. The NVC profiles in Fig. 4f, however, exhibit a maximum drop in NVC of $\Delta_{max}^{dec} = -0.25$ along lines M2 and M3 and increase by $\Delta_{max}^{inc} = +0.2$ along lines M1 and M4. This implies that approximately 80% of the depleted V_o laterally moved with the blunt tip. The strong reduction (improvement) of Δ_{max}^{dec} (Δ_{max}^{inc}) thus confirms an active suppression of the out-of-plane migration and a simultaneous enhancement in the lateral transportation of V_o , during scans with the blunt tip. Overall, the ability to deterministically move V_o constitutes the first experimental demonstration of a controlled manipulation of V_o in an oxide and the resulting two-dimensional spatial modulation.”

Q.3e

Furthermore, to support their proposed mechanism, which is in their words contentious, the authors could provide a number of controlled experiments. For instance, they could scan by applying force only during trace (and not during retrace). This could rule out a simple mechanical dragging of charged adsorbates during scanning if the results were insensitive to whether force was applied from left to right or from right to left (according to their proposed mechanism this should not matter) (Q.3e1). Furthermore, if diffusion was the transport mechanism, then the results should be sensitive to scanning velocity (Q.3e2). Finally, their proposed mechanism should produce a smaller OV enrichment the farther apart from the OV-rich region (Q.3e3).

A. 3e

We would like to thank the reviewer for suggesting these additional experiments for further supporting our proposed mechanism. We used a sharp tip and the 120-uc thick STO film for these additional experiments. We have split our response into following subsections.

A. 3e1

Following the reviewer’s suggestion, we performed mechanical scans by tracing the tip along a predefined path as shown in Figs. R3a-b. During these scans, the tip was traced across the boundary between the OV-rich and pristine regions following left-right-left-right sequence. Note that each horizontal traces are vertically offset by a distance of about $1\mu m$, enabling us separately comparing the OV-redistribution when the tip moves from left-right and right-left. As can be seen from NVC maps in Figs. R3a and 3b and the associated profiles shown in Figs. R3c and 3b, the results are insensitive to whether the tip moves left-right or from right-left.

Regarding the possibility of mechanical dragging of charged adsorbates, we think that this experiment does not provide a clear answer. However, based on the following considerations we believe that the role of charged adsorbates is negligible.

- 1) The time evolution of KPFM signal in Figs. 1c-e of the revised manuscript shows a pronounced thickness dependence, which can only be explained by the diffusion of OVs.
- 2) Earlier works have elaborated that an applied force in the range of $0.5\mu N$ - $1\mu N$ is sufficient for removing charged adsorbates [Nano Lett. 15, 3547 (2015), Proc. Natl. Acad. Sci. U.S.A.111, 6566 (2014)]. In our work, we used much higher forces than this threshold value.
- 3) Charged adsorbates, namely H^+ and OH^- are highly mobile at room temperature—their diffusion coefficient is about $10^{-7} cm^2/s$ [ACS Nano. 8, 6806, (2013)], which is several orders of magnitude higher

Figure R3 a-b, NVC maps after mechanical scans were performed with a load of $6 \mu\text{N}$ across the left **(a)** right **(b)** boundaries between V_o^- -enriched and pristine regions. During these scans, the tip was traced following a predefined defined path, which is marked by small triangles. The start (end) of the trace is marked by the green (red) triangle. **c-d**, NVC profiles along lines V_1 and V_2 on the map in **a** **(c)** and in **b** **(d)**. The NVC profile along the line V_1 (V_2) shows a decrease (increase) in V_o^- concentration inside (outside) the V_o^- -enriched regions. For clarity, we used vertical arrows to mark the increase in vacancy concentration in the lower panel of **c** and **d**. We have used Fig. **R3** as the Supplementary Fig. 9 of the revised manuscript.

than that of OV_s ($= 10^{-17\pm3} \text{ cm}^2/\text{s}$). In the following subsection (**A. 3e2**), we will show that the net increase in NVC outside the OV-rich region depends on the scanning velocity. The result would have been insensitive to the scanning velocity if the tip primarily dragged charged adsorbates during its lateral motion.

A. 3e2

Following the reviewer's suggestion, we have performed two separate experiments by varying the scanning velocity. First, we performed scan within an OV-rich region. Second, we scanned across the boundary between an OV-rich and pristine region.

Figure R4 shows the results of the first experiment. We applied force only during the trace (left-right), and used four different scanning velocities. These scans consist of tracing five lines over a $1 \times 0.5 \mu\text{m}^2$ area. Note that because of the high background, this experiment does not allow us probing the lateral migration of OV_s, and thus just sensitive to the vertical migration of OV_s. Figure R4b plots the background subtracted NVC (ΔNVC) as a function of the scanning velocity. The monotonic increase in

Figure R4 | **a**, The NVC map after mechanical scans were performed with different scanning velocities (0.1-20 $\mu\text{m/s}$) within a OV-rich region. Scans were performed with a load of 6 μN . The force was applied only during the trace, and the tip was lift-off during the retrace. The horizontal arrow marks the trace direction. **b**, Plot of the background subtracted NVC (ΔNVC) as a function of scanning velocity. The solid line is a guide for eyes. Note an increased drop in NVC around the left and right sides, which is particularly discernible for scans performed with velocities of 4 $\mu\text{m/s}$ and 20 $\mu\text{m/s}$. This larger depletion is due to a stronger deformation of the STO surface during the tip's engagement and withdrawal from the STO surface. We have excluded these regions while plotting ΔNVC in (b). We have used Fig. R4 as the Supplementary Fig. 10 of the revised text.

ΔNVC with decreasing velocity suggests that the longer (shorter) the tip stays in contact with the OV-rich surface, the larger (smaller) number of OVs it pushes into the bulk. This result is consistent with our proposal that the depolarisation field beneath the tip causes OVs to diffuse/migrate into the bulk from the surface.

Next, we discuss the results of the second experiment. The scanning parameters of this second experiment were identical as that of the first experiment. However, the scans were performed over a $3 \times 1.5 \mu\text{m}^2$ area and using two different velocities. Figure R5 shows the results from this experiment. NVC profiles in Fig. R5b shows that the slower scan yields a larger decrease (increase) in NVC inside (outside) the OV-rich region. The larger drop in NVC for the smaller scanning velocity is consistent with our observation in Fig. R4. Thus, in line with our previous argument, the larger gain in NVC outside the OV-rich region can be attributed to an enhanced in-plane diffusion/accumulation of OVs around the contact edge of the tip.

It is worth mentioning that in the above experiment (Fig. R5) the force was applied only during the trace. This implies that the contribution of the retrace scan, which is present in data shown in Figs. 2 and 4 of the revised manuscript are absent in Fig. R5. However, Fig. R5b suggests that regardless of the scanning velocity, the fraction of depleted OVs that laterally move is smaller than those migrate into the bulk. This observation is consistent with our proposed model which points to a dominating role of the depolarisation field beneath a sharp tip.

A. 3e3

For illustrating how the OV-enrichment evolves with increasing distance from the OV-rich region, we show in Fig. R6 a part of Fig. R5, which magnifies the region scanned with the velocity of 0.5 $\mu\text{m/s}$. As the reviewer predicted, the NVC profiles in Fig. R6b show that the OV-enrichment gradually decreases with increasing distance from the OV-rich region. We made a similar observation for the scan that was performed with the tip velocity of 20 $\mu\text{m/s}$.

Figure R5 | **a**, The NVC map after mechanical scans were performed with two different scanning velocities (0.5 and 20 μm/s) across the boundary between a V_o²⁻-enriched and pristine regions. Scans were performed with a load of 6 μN. The force was applied only during the trace, and the tip was lift-off during the retrace. The horizontal arrow marks the trace direction. **b**, NVC profiles along lines V₁ and V₂, showing vacancy depletion inside the V_o²⁻-enriched and enrichment outside this region. The corresponding scanning velocities are indicated on the plot. The profiles are averaged over a 0.5 μm wide window. Note the region scanned with the tip velocity of 20 μm/s shows a large increase in NVC at the end of the trace. This is an artefact, caused by a larger deformation of the STO surface during the withdrawal of the tip from the STO surface. We often made similar observations even for an engagement of the tip on the pristine region. For profiling the NVC map in (a), we have thus excluded the left and right ends of the scanned regions. We have used Fig. R5 as the Supplementary Fig. 11 of the revised manuscript.

Figure R6 | **a**, Part of the NVC map shown in Fig. R5. This NVC map magnifies the region, which was scanned with the tip velocity of 0.5 μm/s. **b**, NVC profiles along the black, red, and green colour lines in a. The lines are placed at a distance of about 0.2 μm, 0.5 μm, and 1 μm from the V_o²⁻-enriched region. A gradual decrease in the vacancy concentration as a function of distance from the V_o²⁻-enriched region is clearly observable. We made a similar observation for the area scanned with the tip velocity of 20 μm/s. Note that for this comparison, we have overlapped the background, which uniformly decreases with increasing distance from the V_o²⁻-enriched region. We have included Fig. R6 as the Supplementary Fig. 12 of the revised manuscript.

We have included Figs. R3-R6 in the Supplementary information of the revised manuscript, and added appropriate discussions.

Moreover, we have replaced the text “The simulation results qualitatively explain the characteristics of the mechanical V_o -redistribution in Fig. 2. During a mechanical scan, a spatially extended and strong downward E_z^{dep} field acts over a larger fraction of V_o underneath the tip, which results in a dominant surface-bulk migration. In contrast, a small fraction of V_o becomes effectively trapped within a shallow annular region around the contact edge by the upward E_z^{dep} and inward $E_{x,y}^{dep}$. These trapped vacancies can move laterally with the tip from the V_o -enriched region to the pristine region during the tip’s lateral motion. Therefore, the contrasting roles of the depolarisation field underneath the tip and around the contact edge corroborate both the dominant surface-bulk migration and the lateral motion of V_o with the tip along the surface. In the following, we illustrate that this depolarisation can be tailored in favour of the lateral motion of V_o , which enables controllably manipulating the vacancy distribution.” with the following one in line #216 of the revised manuscript.

“ The simulation results qualitatively explain the characteristics of the mechanical V_o -redistribution in Fig. 2. During a mechanical scan, a spatially extended and strong downward E_z^{dep} field acts over a larger fraction of V_o underneath the tip, which results in a dominant surface-bulk migration. In contrast, a small fraction of V_o becomes effectively trapped within a shallow annular region around the contact edge by the upward E_z^{dep} and inward $E_{x,y}^{dep}$. These trapped vacancies can move laterally with the tip from the V_o -enriched region to the pristine region during the tip’s lateral motion. Therefore, the contrasting roles of the depolarisation field underneath the tip and around the contact edge corroborate both the dominant surface-bulk migration and the relatively weaker lateral motion of V_o with the tip along the surface.

Notably, while scanning, the tip redistributes V_o regardless whether it moves from left-right or right-left. Thus, the mechanical redistribution of V_o should be understood as an average response of V_o to the force applied during the trace and retrace. Moreover, the scanning velocity would influence the V_o -redistribution—longer the tip spends in contact with STO, the larger number of V_o it would redistribute. To check whether these factors could contribute to the weaker lateral motion of V_o , we performed additional experiments, whereby we applied force only during the trace, and varied the scanning velocity (see Supplementary information). These experiments also yielded a weaker lateral motion but a dominant surface-bulk migration of V_o —highlighting the dominating influence of depolarisation field underneath the tip. In the following, we illustrate that the depolarisation field around the tip-STO contact junction can be tailored in favour of the lateral motion of V_o , which enables controllably manipulating the vacancy distribution.”

* A Summary of changes in the revised manuscript

- 1) We removed the word “direct” from line #18 of the revised manuscript.
- 2) We replaced the text “Hence, the argument that the flexoelectric field can act on V_o^{\cdot} is contentious.” with “*Hence, it remains unclear how this flexoelectric field acts on V_o^{\cdot} .*” in line # 54 of the revised manuscript.
- 3) We removed the word “novel” from line #62 of the revised manuscript.
- 4) We replaced the text “As shown in Fig. 1b (upper panel), the accumulation of V_o^{\cdot} on the STO surface locally alters this CPD.” with “*However, as schematically illustrated in Fig. 1b (upper panel), this CPD would change if the STO surface contains V_o^{\cdot} , which can be locally accumulated by an electrical poling.*” in line #76 of the revised manuscript.
- 5) We included the text “*Moreover, the tip bias used during the KPFM measurement has been argued to influence the measured CPD²⁹.*” in line #79 of the revised manuscript.
- 8) We replaced the text “The influence of each factor cannot be unambiguously separated.” with “*The influence of each factor cannot be individually separated, which inhibits a quantitative determination of the vacancy concentration with the KPFM technique.*” in line # 80 of the revised manuscript.
- 9) We replaced the phrase “However, as illustrated in Fig. 1b (lower panel)” with “*Nonetheless, as shown in Fig. 1b (lower panel)*” in line #82 of the revised manuscript.
- 10) We replaced the text “To establish a proof of concept, we characterised the diffusion of V_o^{\cdot} .” with “*To establish a proof of concept, in the following we elaborate on the application of KPFM technique to study the diffusion of V_o^{\cdot} using a 14 and 120-unit cell (uc)-thick STO film.*” in line #84 of the revised manuscript.
- 11) We replaced the text “Figures 1c-d show KPFM images after poling the pristine surface of a 14 and 120-unit cell (uc)-thick STO films with a tip bias of -5 V.” with “*Figures 1c-d show KPFM images after poling the pristine surface of STO films with a tip bias of -5 V.*” in line #86 of the revised manuscript.
- 12) We replaced the text “In contrast, because of diffusion, the vacancy concentration is expected to decrease with a strong thickness-dependent timescale³¹.” with “*In contrast, because of diffusion or surface reaction that enables the recombination of V_o^{\cdot} with oxygen from the ambient, the vacancy concentration is expected to decrease with a pronounced thickness-dependent timescale³².*” in line #91 of the revised manuscript.
- 13) We replaced the text “Therefore, we argue that the surface charging is caused by V_o^{\cdot} , which undergo diffusion with time.” with “*Therefore, we argue that the surface charging is caused by V_o^{\cdot} , which either undergo diffusion or recombine with oxygen from the ambient.*” in line #95 of the revised manuscript.
- 14) We replaced the text “This diffusion process can be quantified in terms of the degree of equilibrium, $S(t)$, which describes the time evolution of the vacancy concentration (see Supplementary Sec. S1). Figure 1e plots $S(t)$ as a function of time. By fitting these data with Fick’s 2nd law of diffusion (solid lines), we obtained the diffusion coefficient $D = 9.4(3) \times 10^{-19}$ and $3.8(2) \times 10^{-18}$ cm²/s for the 14-uc and 120-uc-thick STO film, respectively. These values agree well with that of the bulk value $D_{bulk} \approx 10^{-(17\pm 3)}$ cm²/s (300K-extrapolated)³², which corroborates that the diminishing image contrast in Figs. 1c-d is consistent with the diffusion of V_o^{\cdot} . Furthermore, a reasonable agreement between D and D_{bulk} also validates the conceptual schematic depicted in the lower panel of Fig. 1b. Subsequently, we utilised this correlation between the vacancy concentration and KPFM signal to image the vacancy redistribution under an applied bias and force.” with “*We can distinguish the dominating mechanism that causes the*

vacancy concentration to decay over time by analysing the time-dependency of the degree of equilibrium, $S(t)$ that can be calculated from the KPFM images (see Supplementary Sec. S1). The time-dependency of $S(t)$ describes how the surface equilibrates after the electrical poling. $S(t)$ will exhibit a semi-parabolic (linear) time-dependency if diffusion (surface reaction) is the dominating mechanism^{32,27}. Figure 1e plots $S(t)$ as a function of time. Evidently, the time-dependencies of $S(t)$ are semi-parabolic for both films—implying that the diffusion of V_o^- causes the vacancy concentration to decrease over time.

Arguably, during the poling, the applied tip bias perturbs the equilibrium V_o^- -distribution of the surface and the entire film underneath, which equilibrates through the diffusion of V_o^- . This diffusion occurs both along the out-of-plane and in-plane directions. However, due to the high surface sensitivity of the KPFM technique, the surface-bulk diffusion of V_o^- predominantly affects the time evolution of the KPFM contrast, and thus the time-dependency of $S(t)$. Notably, during this surface-bulk diffusion, the repulsive vacancy-vacancy interaction inhibits V_o^- to migrate independently along the out-of-plane direction³³. Thus, the time-dependency of $S(t)$ effectively describes how the perturbed volume under the poled area equilibrates. Naturally, this volume would be smaller in the thinner STO film, and thus would equilibrate faster (see Supplementary Sec. S1 for a detailed discussion). This explains why the KPFM contrast ($S(t)$) diminishes (grows) more rapidly in the 14-uc thick STO than in the 120-uc thick STO film (Figs. 1c-e).

Following the rationale above, we fit the time evolution of $S(t)$ with Fick's 2nd law of diffusion (solid lines in Fig. 1e). From this fitting, we obtained the diffusion coefficient $D = 9.4(3) \times 10^{-19}$ and $3.8(2) \times 10^{-18}$ cm²/s for the 14-uc and 120-uc-thick STO film, respectively. These values are well in the range of the bulk value $D_{\text{bulk}} \approx 10^{-(1.7 \pm 3)}$ cm²/s (300K-extrapolated)³⁴, which validates the conceptual schematic depicted in the lower panel of Fig. 1b. Subsequently, we utilised this correlation between the vacancy concentration and KPFM signal to image the vacancy redistribution under an applied bias and force.” in line #97 of the revised manuscript.

15) We replaced the text “An identical functional dependence of NVC on the applied bias and force implies that the force functionally acts as a positive bias along M.” with “*This functional analysis suggests that the applied positive bias and force have a same qualitative effect along the lines E and M, respectively.*” in line #146 of the revised manuscript.

16) We added a new text “*Also, the possibility of recombination with oxygen from the ambient during the lateral motion of V_o^- can be ignored based on the following considerations. The influence of the surface reaction process, which could facilitate this recombination is negligible in our film. Furthermore, the use of a grounded during the mechanical scan rules out the bias-induced amplification of this surface reaction process³⁷.*” in line #165 of the revised manuscript.

17) We added a new text “*To understand the mechanical redistribution of V_o^- we first considered two mechanisms—the converse Vegard effect and the flexoelectric effect^{15,16}. Recently, the magnitude of these two effects under an applied force from SPM tip has been compared in a PbTiO₃ (PTO) thin film³⁸. This study suggests that the converse Vegard effect is much weaker than the flexoelectric effect. Notably, both the PTO and STO have comparable flexoelectric and Vegard coefficients^{25,38,39}, which determine the relative contributions of these two effects for a given force. Based on these considerations, we thus conclude that the flexoelectric effect predominantly causes the mechanical redistribution of V_o^- , and the contribution from the converse Vegard effect is marginal.*” in line #179 of the revised manuscript.

18) We replaced the text “To understand the mechanical V_o^- -redistribution, we performed phase-field simulations; whereby we incorporated the flexoelectric effect³⁶ and coupled the time-dependent Ginzburg-Landau and the Nernst-Planck equations^{37,38}” with “*For gaining a mechanistic understanding of this flexoelectric effect-driven V_o^- -redistribution, we performed phase-field simulations; whereby we*

incorporated the flexoelectric effect³⁶ and coupled the time-dependent Ginzburg-Landau and the Nernst-Planck equations^{37,38}.” in line #187 of the revised text.

19) We added a new text *“Since in the simulation the STO film is assumed to be in the paraelectric phase, the stress-gradient-induced polarisation in Fig. 3a purely stems from the flexoelectric effect.”* in line #198 of the revised text.

20) We replaced the text *“The simulation results qualitatively explain the characteristics of the mechanical V_o -redistribution in Fig. 2. During a mechanical scan, a spatially extended and strong downward E_z^{dep} field acts over a larger fraction of V_o underneath the tip, which results in a dominant surface-bulk migration. In contrast, a small fraction of V_o becomes effectively trapped within a shallow annular region around the contact edge by the upward E_z^{dep} and inward $E_{x,y}^{dep}$. These trapped vacancies can move laterally with the tip from the V_o -enriched region to the pristine region during the tip’s lateral motion. Therefore, the contrasting roles of the depolarisation field underneath the tip and around the contact edge corroborate both the dominant surface-bulk migration and the lateral motion of V_o with the tip along the surface. In the following, we illustrate that this depolarisation can be tailored in favour of the lateral motion of V_o , which enables controllably manipulating the vacancy distribution.”* with *“The simulation results qualitatively explain the characteristics of the mechanical V_o -redistribution in Fig. 2. During a mechanical scan, a spatially extended and strong downward E_z^{dep} field acts over a larger fraction of V_o underneath the tip, which results in a dominant surface-bulk migration. In contrast, a small fraction of V_o becomes effectively trapped within a shallow annular region around the contact edge by the upward E_z^{dep} and inward $E_{x,y}^{dep}$. These trapped vacancies can move laterally with the tip from the V_o -enriched region to the pristine region during the tip’s lateral motion. Therefore, the contrasting roles of the depolarisation field underneath the tip and around the contact edge corroborate both the dominant surface-bulk migration and the relatively weaker lateral motion of V_o with the tip along the surface.*

Notably, while scanning, the tip redistributes V_o regardless whether it moves from left-right or right-left. Thus, the mechanical redistribution of V_o should be understood as an average response of V_o to the force applied during the trace and retrace. Moreover, the scanning velocity would influence the V_o -redistribution—longer the tip spends in contact with STO, the larger number of V_o it would redistribute. To check whether these factors could contribute to the weaker lateral motion of V_o , we performed additional experiments, whereby we applied force only during the trace, and varied the scanning velocity (see Supplementary information). These experiments also yielded a weaker lateral motion but a dominant surface-bulk migration of V_o —highlighting the dominating influence of depolarisation field underneath the tip. In the following, we illustrate that the depolarisation field around the tip-STO contact junction can be tailored in favour of the lateral motion of V_o , which enables controllably manipulating the vacancy distribution.” in line # 216 of the revised manuscript.

20) We replaced the word “indenter” with “tip” in the revised manuscript.

21) We replaced the text *“To validate our proposition, we performed experiments with a blunt tip.”* with *“To validate our proposition, we performed experiments with a sharp and blunt tip.”* in line # 250 of the revised manuscript.

22) We replace the text *“Figure 4c shows the NVC map after mechanical scans were performed with this tip at a contact force of 9.5 μ N.”* with *“We used the 120-uc thick STO film in these experiments. Figures 4c-d show the NVC maps after mechanical scans were performed with these tips at a contact force of 9.5 μ N.”* in line #252 of the revised manuscript.

23) We replaced the text *“To characterise this blunt tip-induced V_o -redistribution we profiled the NVC map, as indicated by vertical lines in Fig. 4c. The corresponding NVC profiles are shown in Fig. 4d.*

The overlapping NVC profiles along lines M1/M4 demonstrate that the V_o -redistribution is reproducible. The NVC profiles exhibit a maximum drop in NVC of $\Delta_{\max}^{\text{dec}} = -0.25$ along lines M2 and M3 and increase by $\Delta_{\max}^{\text{inc}} = +0.2$ along lines M1 and M4. This implies that approximately 80% of the depleted V_o laterally moved with the tip. Furthermore, pristine regions across borders are twice as much populous as that in Fig. 2b. These observations confirm an active suppression of the out-of-plane migration and a simultaneous enhancement in the lateral transportation of V_o , as we proposed based on the above simulation. Overall, the ability to deterministically move V_o constitutes the first experimental demonstration of a controlled manipulation of V_o in an oxide and the resulting two-dimensional spatial modulation.” with *“To compare the sharp and blunt tip-induced V_o -redistribution we profiled the NVC maps, as indicated by vertical lines in Figs. 4c-d. Figures 4e-f show the corresponding NVC profiles. The overlapping NVC profiles along lines M1/M4 and M2/M3 demonstrate that the V_o -redistribution is reproducible for both tip geometries. However, the response of V_o to the applied force from these two tips are clearly different. The NVC profiles in Fig. 4e exhibit a maximum drop in NVC of $\Delta_{\max}^{\text{dec}} = -0.75$ along lines M2 and M3 but no appreciable increase in NVC ($\Delta_{\max}^{\text{inc}}$) along lines M1 and M4. This implies that the sharp tip strongly depletes the V_o -enriched regions but barely enriches the pristine regions. This result is in qualitative agreement with that in Fig. 2e, which shows that the fraction of the depleted V_o that laterally move with the tip progressively decreases for applied forces larger than $6\mu\text{N}$. The NVC profiles in Fig. 4f, however, exhibit a maximum drop in NVC of $\Delta_{\max}^{\text{dec}} = -0.25$ along lines M2 and M3 and increase by $\Delta_{\max}^{\text{inc}} = +0.2$ along lines M1 and M4. This implies that approximately 80% of the depleted V_o laterally moved with the blunt tip. The strong reduction (improvement) of $\Delta_{\max}^{\text{dec}}$ ($\Delta_{\max}^{\text{inc}}$) thus confirms an active suppression of the out-of-plane migration and a simultaneous enhancement in the lateral transportation of V_o , during scans with the blunt tip. Overall, the ability to deterministically move V_o constitutes the first experimental demonstration of a controlled manipulation of V_o in an oxide and the resulting two-dimensional spatial modulation.”* in line # 256 of the revised manuscript.

24) We replaced the text “This could enable, using an SPM-based all-in-one platform, the investigation of quantum phase transitions and emergent phenomena in mesoscopic limits³⁸” with *“This could enable, using an SPM-based all-in-one platform, the investigation of mesoscale quantum phenomena in oxides⁴³.”* in line #276 of the revised manuscript.

25) We replaced the text “Therefore, the model can be extended to the study of exotic conducting properties of charged domain walls and morphotropic phase boundaries in ferroelectric oxides^{42,43,44},” with *“Therefore, the model can be extended to the study of emergent problems such as the domain wall conductivity, high electrical conductivity of morphotropic phase boundaries, and leakage current in ferroelectric oxides^{47,48,49,50}.”* in line #290 of the revised manuscript.

26) We corrected minor errors in the Methods section. In line #305 of the revised manuscript, we changed the phrase “lift height of 40 nm” to *“lift height of 30 nm”*. We removed “ $V_{dc} = +3 \text{ V}$ ” from line #306 of the revised manuscript.

Reviewers' comments:

Reviewer #2 (Remarks to the Author):

I appreciate very much a very large amount of work carried out by the authors to address the reviewers' comments. They significantly improved the manuscript by clarifying the confusing or questionable points, discussing more properly the data and adding more information, especially into the supplement. In particular, I appreciate the more detailed discussion of lateral diffusion and recombination of the vacancies as well as a role of the tip-induced polarization. It would be a good idea to add a shorter version of the discussion about the influence of the biased tip on the STO polarization given in the authors reply to the reviewers into the supplementary information. I am also impressed by the additional convincing experiments further proving the modelling and declared concepts. I consider the overall quality of the revised version to be considerably increased meeting the requirements of Nature Communication.

Reviewer #3 (Remarks to the Author):

Having read the author's response to my comments, and those of the other referees, I think that the manuscript is much stronger now. I recommend it for publication provided the authors address one additional point raised by the new control experiments testing the role of charged adsorbates. The authors provide additional evidence sustaining that the role of charged adsorbates is negligible, so this point is clarified, but the new experiments raise a new question. In figure R3, it is apparent that OV enrichment in the pristine region occurs when the tip is scanned from the OV-rich region towards the pristine region, and not when it is scanned from the pristine region towards the OV-rich region. I believe this does not contradict the author's model. However, they claim that "the results are insensitive to whether the tip moves left-right or from right-left", which is not really true according to figure R3. The fact that the results are sensitive to whether the tip moves left-right or right-left remains uncommented but the data is very eloquent. A discussion about this should be included.

Reviewer #2 (Remarks to the Author):

I appreciate very much a very large amount of work carried out by the authors to address the reviewers' comments. They significantly improved the manuscript by clarifying the confusing or questionable points, discussing more properly the data and adding more information, especially into the supplement. In particular, I appreciate the more detailed discussion of lateral diffusion and recombination of the vacancies as well as a role of the tip-induced polarization. **It would be a good idea to add a shorter version of the discussion about the influence of the biased tip on the STO polarization given in the authors reply to the reviewers into the supplementary information (Q. 2).**

I am also impressed by the additional convincing experiments further proving the modelling and declared concepts.

I consider the overall quality of the revised version to be considerably increased meeting the requirements of Nature Communication.

A. 2

We would like to thank the reviewer for the positive evaluation of our revised manuscript. Following the reviewer's suggestion, we have included a brief discussion about the influence of the biased tip on the STO polarization in the Supplementary information of the revised manuscript, please see Supplementary figure 13 and associated discussions.

Reviewer #3 (Remarks to the Author):

Having read the author's response to my comments, and those of the other referees, I think that the manuscript is much stronger now. I recommend it for publication provided the authors address one additional point raised by the new control experiments testing the role of charged adsorbates. The authors provide additional evidence sustaining that the role of charged adsorbates is negligible, so this point is clarified, but the new experiments raise a new question. In figure R3, it is apparent that OV enrichment in the pristine region occurs when the tip is scanned from the OV-rich region towards the pristine region, and not when it is scanned from the pristine region towards the OV-rich region. I believe this does not contradict the author's model. However, they claim that "the results are insensitive to whether the tip moves left-right or from right-left", which is not really true according to figure R3. The fact that the results are sensitive to whether the tip moves left-right or right-left remains uncommented but the data is very eloquent. A discussion about this should be included (Q. 3).

A. 3

We would like to thank the reviewer for recommending the revised manuscript for publication after addressing their concern.

In our previous response letter, we intended to imply that irrespective of the scan direction the tip always depletes the OV-rich region and enriches the pristine region whenever it is scanned from the former towards the latter region. We are sorry for not being able to convey our message clearly. We agree with the reviewer's observation that OV-enrichment in the pristine region does not occur when the tip is scanned from the pristine region towards the OV-rich region. Notably, this result is also consistent with the one shown in Supplementary figure 5 of the revised manuscript, which shows that the OV-concentration barely changes when the pristine region is raster scanned with the tip (please see Supplementary Note 2 for a detailed discussion). These results primarily imply that the mechanical scan does not produce OVs on the STO surface, and also rule out the contact electrification or triboelectric charging of the STO surface during the scan. These observations further strengthen our claim: the enrichment of the pristine region during a mechanical scan is purely caused by OVs that laterally move with the tip from the OV-rich to the pristine region.

To comply with the reviewer's suggestion, we have now explicitly mentioned these aforementioned points in Supplementary Note 2 as well as while discussing Supplementary figure 9 (figure R3 of the previous response letter) of the revised manuscript.

Additionally, we replaced the text "Figure 2e shows the NVC with the background (NVC at 0.6 μ N) subtracted (i.e., Δ NVC) along lines M and M_L . Notably, the background within the pristine region does not vary during the mechanical scan, so it can be treated as constant (see Supplementary Sec. S2)." with "*Additional experiments elaborate that the mechanical scan does not alter the background within the pristine region, and thus rule out the formation of V_o^- and triboelectric charging of the STO surface during the mechanical scan (see Supplementary Note 2). To quantify the lateral motion of V_o^- , in Fig. 2e we therefore show the NVC with the background (NVC at 0.6 μ N) subtracted (i.e., Δ NVC) along lines M and M_L .*" in line # 162 of the revised manuscript.

REVIEWERS' COMMENTS:

Reviewer #3 (Remarks to the Author):

I am satisfied with the authors' response. I recommend the manuscript for publication in Nature Communications.